# RNA degradation triggered by decapping is largely independent of initial deadenylation

Léna Audebert [ID] [1,2,7], Frank Feuerbach[1], Mostafa Zedan [ID] [3], Alexandra P Schürch[3], Laurence Decourty[1,4], Abdelkader Namane[1], Emmanuelle Permal [ID] [1,5], Karsten Weis [ID] [3], Gwenaël Badis [ID] [1,6] & Cosmin Saveanu [ID] [1,4 ✉]

## Abstract

**RNA stability, important for eukaryotic gene expression, is thought to depend on deadenylation rates, with shortened poly(A) tails triggering decapping and 5′ to 3′ degradation. In contrast to this view, recent large-scale studies indicate that the most unstable mRNAs have, on average, long poly(A) tails. To clarify the role of deadenylation in mRNA decay, we first modeled mRNA poly(A) tail kinetics and mRNA stability in yeast. Independent of deadenylation rates, differences in mRNA decapping rates alone were sufficient to explain current large-scale results. To test the hypothesis that deadenylation and decapping are uncoupled, we used rapid depletion of decapping and deadenylation enzymes and measured changes in mRNA levels, poly(A) length and stability, both transcriptome-wide and with individual reporters. These experiments revealed that perturbations in poly(A) tail length did not correlate with variations in mRNA stability. Thus, while deadenylation may be critical for specific regulatory mechanisms, our results suggest that for most yeast mRNAs, it is not critical for mRNA decapping and degradation.**

**Keywords** mRNA Degradation; Deadenylation; Decapping; *Saccharomyces cerevisiae*; Modeling
**Subject Categories** Computational Biology; RNA Biology; Translation & Protein Quality

## Introduction

In addition to RNA production, RNA degradation is a key part of gene expression and its regulation. Yet, it is still difficult to predict from sequence alone which mRNAs will be degraded slowly and which will be degraded more rapidly. However, conceptual and mechanistic advances over the last decade tend to indicate that translation plays a crucial role in initiating mRNA degradation and in determining the fate of mRNAs (reviewed in Wu and Bazzini, 2023). While codon usage, ribosome occupancy and stalling have been shown to affect mRNA half-life in a wide range of species, from yeast (Presnyak et al, 2015; Chan et al, 2018) to zebrafish (Mishima and Tomari, 2016) and mammals (Wu et al, 2019), the mechanisms by which ribosomal activity is linked to RNA degradation remain unclear.

Deadenylation by the highly conserved Ccr4-Not complex (reviewed in Raisch and Valkov, 2022; Collart, 2016) has been proposed to be critical for mRNA decay and link stalled translation to mRNA degradation. A component of Ccr4-Not can sense the presence of an empty E site of a translating ribosome and recruit the deadenylation complex to the mRNA (Absmeier et al, 2023; Buschauer et al, 2020). Deadenylation leads to short poly(A) tails that can be recognized by the Lsm1-7/Pat1 complex, which in turn leads to the recruitment of the decapping enzyme to remove the 5′ cap, initiating degradation via the 5′ to 3′ pathway (Tharun and Parker, 2001b; Bouveret et al, 2000). This model was established using pulse-chase experiments with reporter mRNAs for which poly(A) tail shortening was found to occur prior to mRNA degradation. Moreover, the deadenylation rate was found to be faster for an unstable mRNA than a stable one (Muhlrad et al, 1994; Decker and Parker, 1993). Together with results obtained in mammalian cells in culture (Yamashita et al, 2005), these experiments contributed to the prevailing view that a central mRNA degradation mechanism is deadenylation-dependent, starting with an essential and limiting deadenylation step, required for the induction of mRNA decapping. Its general relevance however remains unclear, as subsequent studies have identified a specificity of the Lsm1-7 effects for specific classes of transcripts (He et al, 2018). Furthermore, the Lsm1-7 complex was shown to have a protective effect against RNA degradation under stress conditions, such as nitrogen starvation (Gatica et al, 2019). It is thus therefore unclear whether the sequence of events starting with deadenylation and ending with decapping is a requirement for most mRNAs, or whether it is specific for some transcripts.

A group of transcripts that do not require deadenylation for their degradation are those targeted by nonsense-mediated mRNA decay (NMD). In this translation-dependent mRNA degradation

[1]Institut Pasteur, Université Paris Cité, CNRS UMR3525, Genetics of Macromolecular Interactions, F-75015 Paris, France. [2]Sorbonne Université, Collège doctoral, F75005 Paris, France. [3]Department of Biology, Institute of Biochemistry, ETH Zurich, Zurich, Switzerland. [4]Institut Pasteur, Université Paris Cité, RNA Biology of Fungal Pathogens, F-75015 Paris, France. [5]Institut Pasteur, Université Paris Cité, Bioinformatics and Biostatistics Hub, F-75015 Paris, France. [6]Institut de Biologie de l'Ecole Normale Supérieure (IBENS), Ecole Normale Supérieure, CNRS, INSERM, PSL Research University, 46 rue d'Ulm, 75005 Paris, France. [7]Present address: Department of Microbiology and Molecular Medicine, University of Geneva, Geneva, Switzerland. ✉E-mail: cosmin.saveanu@pasteur.fr

pathway, decapping activation is independent of deadenylation (Muhlrad and Parker, 1994) and influenced by the frequency at which translating ribosomes encounter a premature termination codon (PTC), as measured by in vivo single molecule imaging (Hoek et al, 2019). Decapping activation occurs probably at highly variable rates even for mRNAs that are not targets of NMD, as suggested by kinetic modeling of experimental RNA decay experiments in yeast (Cao and Parker, 2001) or mammalian cells (Eisen et al, 2020). These analyses suggest that once an mRNA is deadenylated, its oligoadenylated state is degraded at highly variable speeds, that can vary by a factor of 1000.

Since it has been experimentally difficult to manipulate deadenylation rates, it has remained unclear if deadenylation is causally linked to mRNA degradation and particularly through activation of decapping. Intriguingly, several large-scale measurements in multiple organisms, including *C. elegans* (Lima et al, 2017) or *A. thaliana* (Jia et al, 2022) have now reavealed that the most stable mRNAs have on average short poly(A) tails at steady state. The anti-correlation between mRNA stability and the poly(A) tail length indicates that deadenylation might not necessarily be a causal element in RNA degradation.

To clarify the importance of deadenylation speed in mRNA degradation, we performed a re-evaluation of RNA degradation models and were able to predict the results of experimental perturbation of poly(A) tail length and mRNA stability. We tested the predictions of these models in *S. cerevisiae* by inactivation of deadenylation and decapping and measured poly(A) tails, mRNA levels and mRNA stability on a large-scale and with specific reporters. Our experimental and computational results suggest that deadenylation speed is not a limiting step for mRNA degradation in yeast. The obtained results are compatible with a model in which deadenylation speed and decapping activation are independent events.

## Results

We became interested in the relationship between poly(A) tail size and mRNA degradation when analyzing the proteins associated with NMD complexes, in particular Upf1 (Dehecq et al, 2018). These complexes contain both Pab1, a protein that efficiently binds long poly(A) tails (Schäfer et al, 2019) and Lsm1, a component of the Lsm1-7/Pat1 complex, which binds preferentially to short poly(A), oligoadenylated, RNAs (Chowdhury et al, 2014; Tharun and Parker, 2001a). We characterized the protein composition of complexes associated with Pab1 (Fig. 1A), Lsm1, Pat1, Lsm7, and Dhh1 (Fig. EV1A–D) and found that, in every case, Upf1 was among the enriched proteins (Dataset EV1). The results were specific, since, for example, Lsm2, present in most of the purified complexes, was absent from Pab1-TAP (Fig. EV1E). The association of Pab1-HA with Upf1 was sensitive to a nuclease treatment (Fig. 1B), supporting the hypothesis that the observed interaction is largely mediated by RNA. Altogether, these results suggest that Upf1 is present in several distinct RNA-protein complexes together with different protein partners.

To examine whether different populations of RNA molecules are present together with Upf1 in Lsm1-7/Pat1 and Pab1-associated mRNPs, we sequenced the RNAs that co-purified with Lsm1-TAP, Pab1-TAP, and Upf1-TAP. A purification of Rpl16a-TAP was done

in parallel, as it allows an estimation of the levels of RNA bound to ribosomes (Halbeisen et al, 2009). With the exception of the Lsm1-TAP purification, the enrichment profiles for RNA with the tagged proteins, were different from a control purification performed with a strain that did not express any tagged protein. We thus focused on the results obtained with purifications of Upf1, Rpl16a, and Pab1.

To validate our results, we analyzed first a sub-category of yeast RNAs, the Xrn1-dependent unstable transcripts (XUT, van Dijk et al, 2011) and the stable unannotated transcripts (SUT, Xu et al, 2009) that are known to be unstable and frequently targets of the NMD pathway (Malabat et al, 2015). As anticipated, we observed a significant enrichment of XUT/SUT transcripts in the Upf1 purification, while XUT/SUT RNAs were strongly depleted from the ribosome associated fraction (Fig. 1C). This depletion was expected, since XUT/SUT only contain spurious short coding sequences and, like other NMD substrates, are probably at most associated with a single translating ribosome (Heyer and Moore, 2016). Surprisingly, XUT/SUT RNAs were enriched in the Pab1 associated fraction. To confirm this observation, we also analyzed the enrichment of a different type of NMD substrates, the pre-mRNAs of ribosomal protein genes, and compared intron-containing transcripts with spliced ones. As observed for the XUT/SUT RNAs, pre-mRNAs were significantly depleted from the ribosome associated fraction, strongly enriched in association with Upf1, but also enriched in the Pab1 bound RNA population (Fig. EV2A, example in Fig. EV2B, data in Dataset EV2). The relative enrichment of the pre-RPL28 NMD sensitive transcript in Upf1-TAP and Pab1-TAP purifications in comparison with the spliced RPL28 mRNA was confirmed by RT-qPCR (Fig. 1D).

To investigate the length of the poly(A) tails of transcripts enriched in the Pab1 associated fraction, we used the ePAT method (Jänicke et al, 2012). As the pre-RPL28 RNA shares the 3′ end with the mature RPL28 mRNA, we chose to examine CCW22, a transcript that has an early stop codon making it a good substrate for NMD, and was associated with Pab1 and Upf1 (Fig. EV2C for a visualization of the reads distribution). The poly(A) tail signal was more extended in the fraction associated with Pab1 compared to total RNA (Fig. EV2D) but the interpretation of this result was complicated by the presence of two transcript ends for CCW22, as evidenced by the TVN-PAT profile (Fig. EV2D, top left). We therefore analyzed another transcript, HHT2, which has short average poly(A) tails and was associated with Pab1 and Upf1 (Fig. EV2C), even if it is not known to be an NMD substrate. Long poly(A) species for HHT2 were enriched in the Pab1-associated fraction (Fig. EV2D, bottom right). Pab1 purification thus allows the enrichment of transcripts with longer poly(A) tails and, in the particular case of HHT2, the detection of an extended poly(A) transcript population that is not visible in the total RNA extract.

Our results showed that Pab1 was associated with unstable transcripts that are also bound by NMD factors. This probably reflects their fast degradation through NMD, a deadenylation-independent mechanism. Since rapid degradation does not allow deadenylation to occur, long poly(A) tails of NMD substrates are here a marker of RNA instability. Interestingly, and independent of their NMD status, transcripts with known long average poly(A) tails (Subtelny et al, 2014) were enriched in the Pab1-associated fraction (Fig. EV2E). We conclude that Pab1 preferentially binds

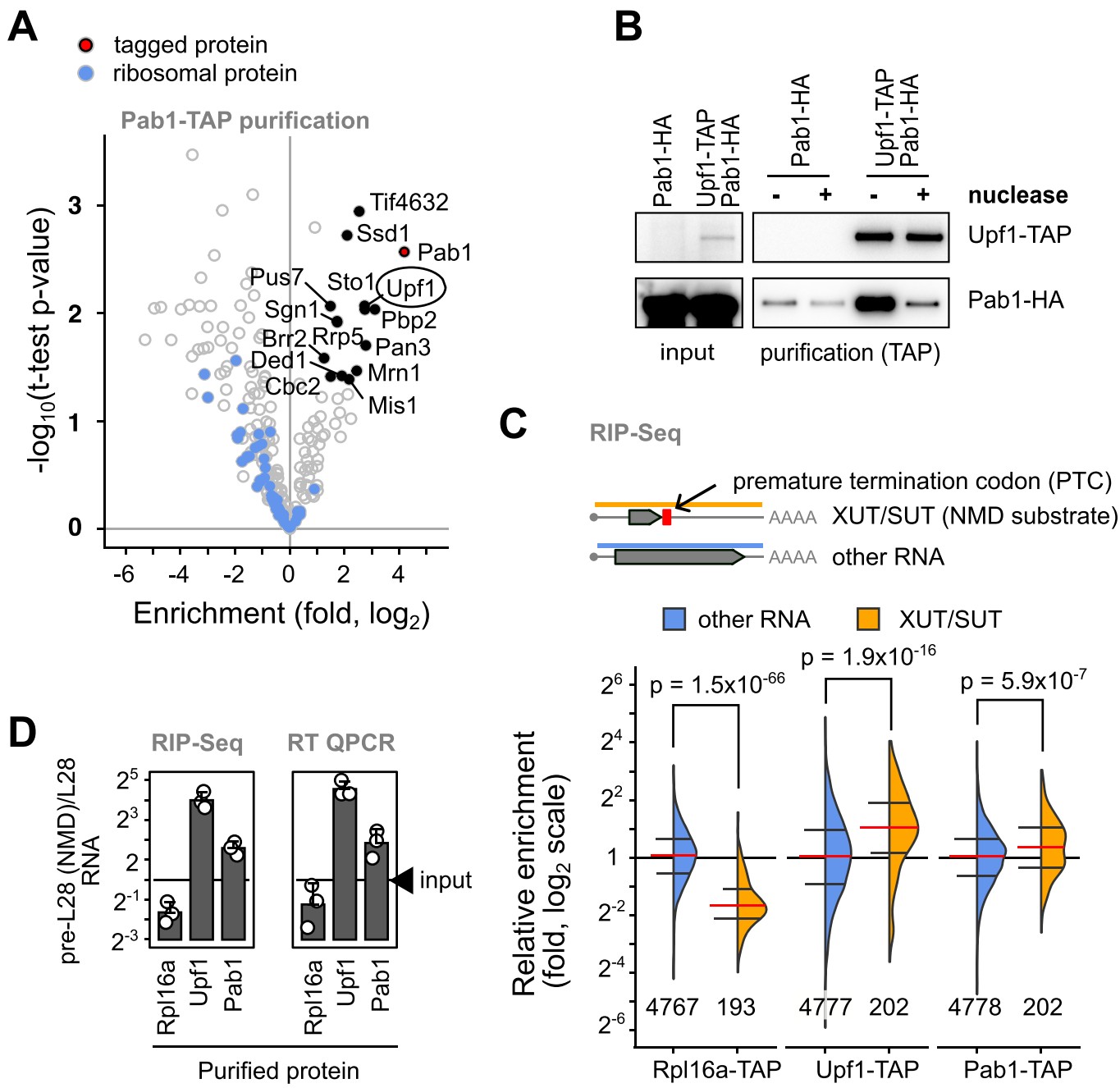

**Figure 1. NMD substrates have long poly(A) tails and are associated with Pab1.**

(A) Enrichment of proteins in association with Pab1-TAP was estimated by quantitative mass spectrometry for identified proteins in comparison with their abundance in a total protein extract. Labeled proteins correspond to those for which at least three independent measures were available, proteins were enriched by a factor higher than 2 and were robustly detected, as judged by a *p*-value lower than 0.05 for a t-test (null hypothesis). Blue dots indicate ribosomal proteins. (B) The interaction between Pab1 and Upf1 was tested by co-purification of Pab1-HA with Upf1-TAP in the presence or absence of micrococcal nuclease treatment. (C) Relative enrichment of NMD substrates, annotated as "XUT/SUT", in association with Rpl16a-TAP, Upf1-TAP, and Pab1-TAP, showed the preference of Upf1 and Pab1 to this unstable RNA population (orange), in comparison with other cellular RNA (blue). The red horizontal line indicates the median of the enrichment values, with the first and third quartiles indicated by horizontal black lines. The indicated *p*-values correspond to a Welch two sample t-test, N is the number of measured transcripts in each category. (D) Validation of RIP-Seq experiments by independent RT-qPCR tests of the relative enrichment of a known NMD substrate, the unspliced pre-mRNA for RPL28 and its non-NMD equivalent, the spliced mRNA. The ratio between the two forms of RNA were compared to the ratio in the total RNA sample (input). Source data are available online for this figure.

long poly(A) tail RNAs and that the association of an mRNA with Pab1 can be considered as a marker of its instability.

The unexpected finding that Pab1 binds unstable RNAs with relatively long poly(A) tails, led us to explore the relationship between RNA half-life and poly(A) tail length from published large-scale data sets. Poly(A) tail length was strongly inversely correlated with RNA stability (Fig. 2A,B; Appendix Fig. S1A,B) as previously reported (Chan et al, 2018; Lima et al, 2017). The observed correlations were independent of the method used for measuring RNA half-life or poly(A) tail length. If deadenylation speed dictates RNA stability, changes in deadenylation activity should be reflected in predictable changes in poly(A) tail lengths for both stable and unstable RNA. To test this hypothesis, we analyzed published changes in poly(A) tail length in strains lacking Pan2 and Ccr4 deadenylases (Tudek et al, 2021). Most stable 25% RNAs and most unstable 25% RNAs showed different relative changes in poly(A) tail length, with a higher relative increase in the poly(A) tail length for relatively stable transcripts in one case, deletion of PAN2, and with a lower effect on this class of transcripts in the absence of CCR4 (Appendix Fig. S1C). Thus, slowing down deadenylation led to changes in poly(A) tail length that were not clearly linked to RNA stability, questioning the impact of deadenylation speed on RNA degradation rates in yeast.

Previous analyses of RNA degradation assumed that deadenylation is the critical step in RNA degradation, and that RNAs are rapidly degraded once a poly(A) tail, of about 10 to 12 nucleotides is reached (Muhlrad et al, 1994; Cao and Parker, 2001). We wondered how well this deadenylation-dependent model for RNA degradation fits the short poly(A) tail length observed for the most stable mRNAs. To this end, we used a kinetic model for RNA degradation inspired by previous work (Cao and Parker, 2001). Three poly(A) species were considered for each RNA, starting with long poly(A) tail RNA (PA$_l$), which is transformed to intermediate (PA$_m$) and short tail RNA (PA$_s$) at a speed proportional to a constant related with deadenylation rate, k$_A$ (Fig. 2C). Oligoadenylated RNA is finally degraded with a first-order constant k$_D$. The functions that express the concentration of each RNA species during time were obtained by solving the set of differential equations for the presented model (Appendix Fig. S1E). A deadenylation-independent model was also considered, in which degradation can also occur for the longer poly(A) tail species (Fig. 2C; Appendix Fig. S1F for the corresponding equations). These models were equally employed within *Tellurium*, a modeling and simulation software package (Choi et al, 2018) using the representation depicted in Appendix Fig. S1D. The computed steady-state levels for RNAs of different poly(A) tail length were used as a starting point in simulated "chase" experiments, to calculate both the variation in average poly(A) tails and the amount of remaining RNA (Fig. 2D, for an example). The values of the rate constants used in these simulations were in the range of those estimated previously from experimental and kinetic modeling results (Cao and Parker, 2001), even if we are aware that measured in vitro decapping rates can be 10 to 50 times higher (Wurm et al, 2017).

To understand the impact of deadenylation and decapping speeds on simulated half-life and poly(A) tail distribution, we varied both k$_A$ and k$_D$ values. For example, for a constant k$_D$ of 0.5 min$^{-1}$, similar in size to the one previously estimated from experimental and modeling data (Cao and Parker, 2001), changing

k$_A$ values by a factor of 25 led to a shift in the half-life of the simulated RNA from 5 to 120 min (Fig. 2E, upper line). With arbitrary values for the three poly(A) species of 70 (PA$_l$), 40 (PA$_m$), and 10 (PA$_s$) nucleotides (Fig. 2C), we computed average sizes of the poly(A) tails at steady-state in each condition. The obtained values ranged from 46 nucleotides for the fastest decaying RNA to 55 nucleotides for the slowest ones (Fig. 2E, upper line). This result is in opposition with the experimentally observed long poly(A) tails for unstable mRNAs (Fig. 2A,B). To solve this conundrum we tested the effect of changing the degradation rate of oligoadenylated RNA, k$_D$. As expected, decreasing k$_D$ by a factor of 5, from 0.5 min$^{-1}$ to 0.1 min$^{-1}$ led to an increase in the RNA half-life from 5 to 9 min and a decrease in the average length of poly(A) tails at steady-state from 46 to 30 nucleotides (Fig. 2E). Further decreasing k$_D$ led to an increase in RNA half-life from 9 to 35 min, and a decrease in average poly(A) tail size from 30 to 16 nucleotides (Fig. 2E). Thus, k$_D$ changes, rather than deadenylation speed might be responsible for the observed correlation between poly(A) size and RNA instability (Fig. 2A,B).

An alternative explanation for the presence of long poly(A) tails for unstable mRNA is that mRNA degradation follows a deadenylation-independent model in which deadenylation and decapping are uncoupled. Such a model involves the addition of just two additional degradation steps (Fig. 2C). Increasing k$_D$ values in this model led to shorter RNA half-lives, with a range of average poly(A) tail lengths that was dependent on the simulated k$_A$ (Fig. 2F).

Thus, the inverse correlation between average poly(A) tail length and RNA half-life (Fig. 2A,B) can only be explained by the classical deadenylation-dependent model when the limiting step for RNA degradation is not deadenylation speed (k$_A$) but rather the degradation of the oligoadenylated RNA species (k$_D$). Alternatively, the observed anti-correlation can also be explained by a model in which decapping can occur even on long poly(A) tail RNAs (Fig. 2C), as it happens for NMD substrates. The two modes of RNA degradation are not mutually exclusive but can be experimentally differentiated by modulating deadenylation or decapping.

Slowing down deadenylation is expected to lead first to the accumulation of unstable RNAs in the classical model and to have no impact if initiation of RNA degradation is not dependent on prior deadenylation (Appendix Fig. S2A,B). By contrast, decapping inhibition should lead to an initial increase in the levels of unstable RNA in both models (Appendix Fig. S2C,D). To test these predictions, we set up conditions in which the major yeast deadenylases Ccr4 and Pop2/Caf1, or the decapping enzyme Dcp2, can be rapidly depleted. The experimental system (Fig. EV3A) is based on the fusion of the protein of interest with an auxin-inducible degron (AID) domain recognized by the auxin-sensitive *O. sativa* TIR1 (Nishimura et al, 2009). The expression of TIR1 was induced using the Z3EV system (Ohira et al, 2017), with a β-estradiol treatment only when the degradation of the protein of interest was required. Addition of indole-3-acetic acid (IAA) auxin together with β-estradiol triggered proteasome-degradation of AID-proteins, as seen by the rapid decrease in the levels of Ccr4, Pop2, or both proteins (Fig. 3A). This decrease in protein levels was accompanied by a slow-growth phenotype, only visible in the presence of IAA and β-estradiol (Fig. EV3B). The growth defect of degron strains could be reverted in presence of a plasmid

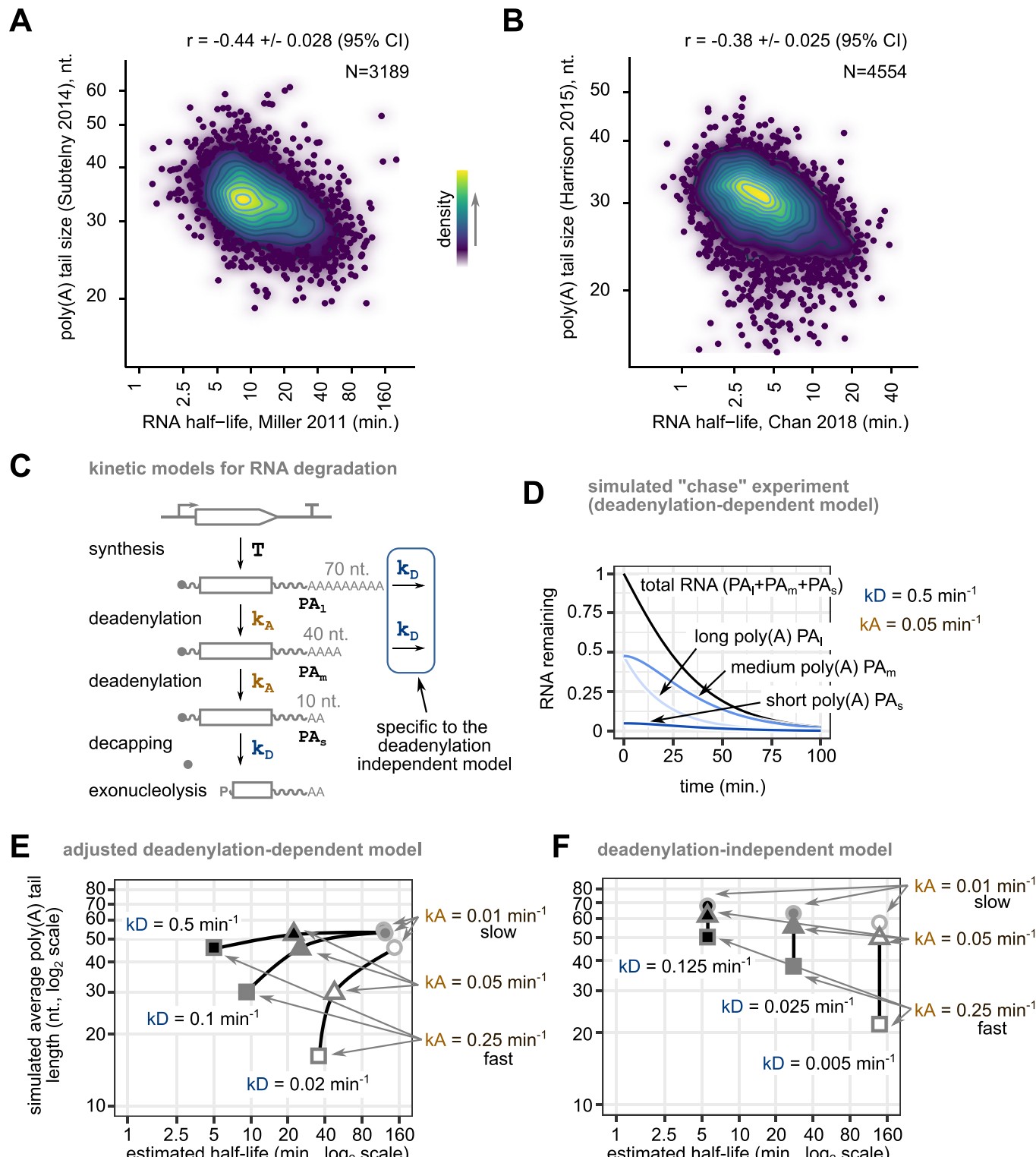

carrying the corresponding deadenylase (Fig. EV3C). Expression of either CCR4 or POP2 could not complement the growth defect of the double degron strain, as expected (Fig. EV3D). As a further validation of these complementation results, we tested the poly(A) status of an endogenous transcript, HHT2, by ePAT. We observed

that the increase in poly(A) length observed when Ccr4 was depleted could be reverted when CCR4 was expressed, but not when a catalytically inactive version of the enzyme, E556A (Chen et al, 2002), was used (Fig. EV3E,F). The shift in poly(A) tail length for HHT2 was more modest when POP2 was depleted, and the

◀ **Figure 2.  The presence of long poly(A) tails on unstable mRNAs can be explained by two RNA degradation mechanisms.**

(A) Average poly(A) tail length shows a negative correlation with RNA half-life. Published poly(A) tail size data (log2, Subtelny et al, 2014), were represented on the vertical axis as a function of estimated RNA half-life, horizontal axis (log2, Miller et al, 2011). The Pearson product moment correlation coefficient and its 95% confidence interval are displayed on the graph. (B) Similar to panel (A), but with data for poly(A) size and RNA half-life from other publications (Harrison et al, 2015; Chan et al, 2018). (C) Depiction of two models for RNA degradation, one in which deadenylation is required to induce decapping, and another in which decapping can occur on RNAs with any size of poly(A) tail. $k_A$ is a kinetic constant for deadenylation, $k_D$ represents a decapping rate constant and "T" indicates all the processes leading to the generation of cytoplasmic mRNA. (D) Example of simulated RNA degradation curve for the deadenylation-dependent kinetic model, starting from a steady-state situation described by a decapping constant of 0.5 min$^{-1}$ and a deadenylation constant of 0.05 min$^{-1}$. The relative variation of the amounts of the three forms of RNA is indicated. (E) Examples of simulated half-life values and average poly(A) tail length for the deadenylation-dependent model for RNAs with three values of $k_D$ (low, white, intermediate, gray, and high, black) in combination with three values of $k_A$ (low, circles, intermediate, triangles, and high, squares). Values of poly(A) for the three species were arbitrarily set to 70, 40, and 10 nucleotides to calculate averages. Both axes use a logarithmic scale. (F) Similar to (E), but for the deadenylation-independent kinetic model. The $k_D$ values were adjusted to obtain half-life and poly(A) size values compatible with published results.

Pop2(D310A) mutant, supposed to affect the enzymatic activity of Pop2, was as effective as the wild type in reverting the poly(A) and growth phenotype (Fig. EV3C,E,G). It is possible that the D310A change is not sufficient to completely inactivate the deadenylation activity of Pop2, as demonstrated in vitro (Ye et al, 2021). Alternatively, the presence of the protein is sufficient to activate the deadenylase activity of Ccr4, as previously suggested (Tucker et al, 2002). In conclusion, the degron system allow rapid depletion of the major deadenylases, with an impact on the poly(A) length of an endogenous transcript.

For a global view of poly(A) tail length and transcript level changes we used Nanopore sequencing (Tudek et al, 2021; Ohira et al, 2017) in strains depleted for the two deadenylases of the Ccr4-Not complex or for Dcp2. Depletion of these proteins led to global changes in poly(A) tail length. As expected, mRNA poly(A) tails became globally longer when both deadenylases were depleted, with an average shift from 42.5 to 52 nucleotides (Fig. 3B). Depletion of Dcp2 had an opposite effect, with a decrease of the average poly(A) tail length to 36.3 nucleotides. This shift was expected if Dcp2 depletion stabilizes mRNAs leaving more time for deadenylation to occur. To judge the impact of poly(A) tail modifications on RNA levels we looked at how transcript poly(A) tail length changed in the degron strains in relation with RNA amounts (Dataset EV3). We found that an increase in mRNA levels was accompanied by a decrease in the size of the poly(A) tails when Dcp2 was depleted (Fig. 3C). Such correlated changes were not observed in the strains depleted for the Ccr4 and Pop2 deadenylases (Fig. 3D), even though the poly(A) tail length was clearly and globally increased. Thus, perturbation of deadenylation or decapping led to results that are compatible with an uncoupling between these processes during the degradation of unstable RNAs in yeast.

Having established that rapid depletion of decapping or deadenylation enzymes had a major impact on the steady-state poly(A) tail length of yeast transcripts, we directly measured the effect of changing deadenylation speed on mRNA half-life. To this end, we performed SLAM-seq experiments (Herzog et al, 2017) in which 4-thiouracil is partially incorporated into newly synthesized mRNA and its levels of mRNA modification are followed over time in a chase experiment in a medium containing an excess of uracil (Fig. 4A). Incorporation of the modified nucleotide into the mRNA can be detected by DNA sequencing after reverse transcription, during which modified 4-thiouracil is misread, leading to a T to C conversion wherever 4-thiouracil was present in the mRNA. First-order decay kinetics applied to T to C conversion events allowed us to obtain half-life estimates for 3383 transcripts (Dataset EV4). The

obtained half-life estimates for the degron strains in the absence of IAA and β-estradiol correlated well with previous half-life measurements obtained with a different detection method (Chan et al, 2018) (Fig. 4B). Our results also correlated well to measurements performed by a regulated promoter assay method (Baudrimont et al, 2017) (Appendix Fig. S3A) and with recent SLAM-seq results (Alalam et al, 2022) (Appendix Fig. S3B).

Interestingly, changes in the mRNA half-life following Dcp2 or Ccr4 and Pop2 depletion were not correlated (Fig. 4C) although both treatments led to a stabilization of a fraction of transcripts. Several transcripts, as illustrated for ADA2 or SRO9, were stabilized only when Dcp2 was depleted (Fig. 4D,E), while others, such as SNP1, showed also a partial stabilization following depletion of Ccr4/Pop2 (Fig. 4F). Finally, some transcripts, such as PIR1, were not sensitive to Dcp2 depletion but were stabilized by depletion of deadenylases (Fig. 4G). Overall, the lack of correlation between the effects of depleting Dcp2 or Ccr4/Pop2 suggests that there is no causal effect linking deadenylation speed with triggering mRNA decapping and subsequent degradation.

One of the limitations of the SLAM seq experiments is that they do not allow a concomitant estimation of the dynamics of both mRNA decay and poly(A) tail length. To compare RNAs with different stabilities, in a context where NMD does not play a role, we took advantage of the previous observation that a coding sequence bias affects mRNA degradation (Presnyak et al, 2015; Herrick et al, 1990) and built two reporter RNAs with HIS3 coding sequences that were either optimal (OPT-HIS3) or non-optimal (non-OPT-HIS3) in terms of codon usage. A tetOFF system was used to turn off reporter synthesis following the addition of doxycycline (Fig. 5A). We measured half-lives of 16 min for OPT-HIS3 and 8 min for the non-OPT-HIS3 reporter (Fig. EV5A). The reporters, particularly the non-OPT-HIS3 mRNA, were stabilized by the addition of the translation inhibitor cycloheximide (Fig. EV4A) and were not affected by depletion of Upf1, which inhibits NMD (Fig. EV4B,C).

Next, we tested the levels of our mRNA reporters at different time points after addition of doxycyline in a strain depleted for Dcp2. The half-life of both reporters doubled in these conditions (Figs. 5B,C and EV5A,B). Intriguingly, the measured half-lives of the non-OPT-HIS3 and OPT-HIS3 mRNA were not affected in the degron strains after depletion of Ccr4, Pop2 or both deadenylases together (Figs. 5D, EV4D,E and EV5C–E). These results were in agreement with the lack of global mRNA level changes upon deadenylase depletion despite the upshift of poly(A) tail lengths, as measured by Nanopore sequencing (Fig. 3B,D).

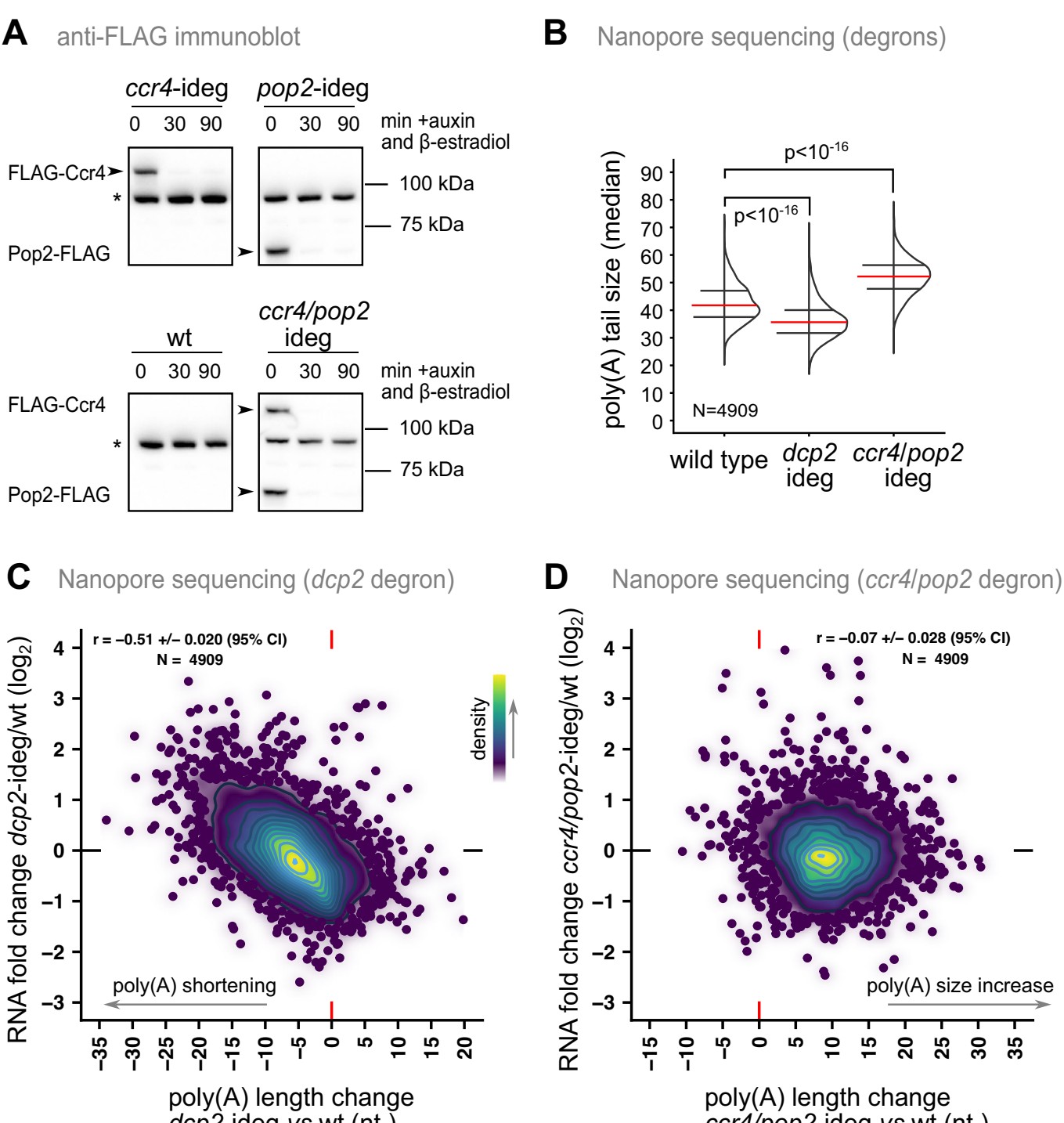

**Figure 3. Poly(A) tail changes when decapping or deadenylation are inactivated suggest a deadenylation-independent model for RNA degradation.**

(A) Immunoblot estimation of the decrease in the levels of deadenylases targeted by the inducible degron system. Protein degradation was induced by addition of β-estradiol (1 μM) and IAA (100 μM) and total protein extracts were tested at the indicated time points with anti-FLAG antibodies. The asterisk indicates a yeast protein that is detected by the antibodies in all samples. (B) Comparison of the global distribution of median poly(A) tails for mRNAs as detected by Nanopore sequencing in strains depleted for Dcp2 or Ccr4 and Pop2 by treatment with β-estradiol and IAA for 1 h in liquid medium. The results represent the average of two independent experiments. Red lines correspond to medians of the distributions, with black lines indicating the first and the third quartiles. Indicated *p*-values correspond to a paired Wilcoxon signed rank test with continuity correction for the null hypothesis. (C) Relationship between the measured changes in poly(A) tail length in Dcp2-depleted cells and the increase in the levels of the corresponding mRNAs, compared with a wild type strain. The Pearson product moment correlation coefficient and its 95% confidence interval are displayed on the graph. (D) Similar to (C) for the strain depleted for the Ccr4 and Pop4 deadenylases. Source data are available online for this figure.

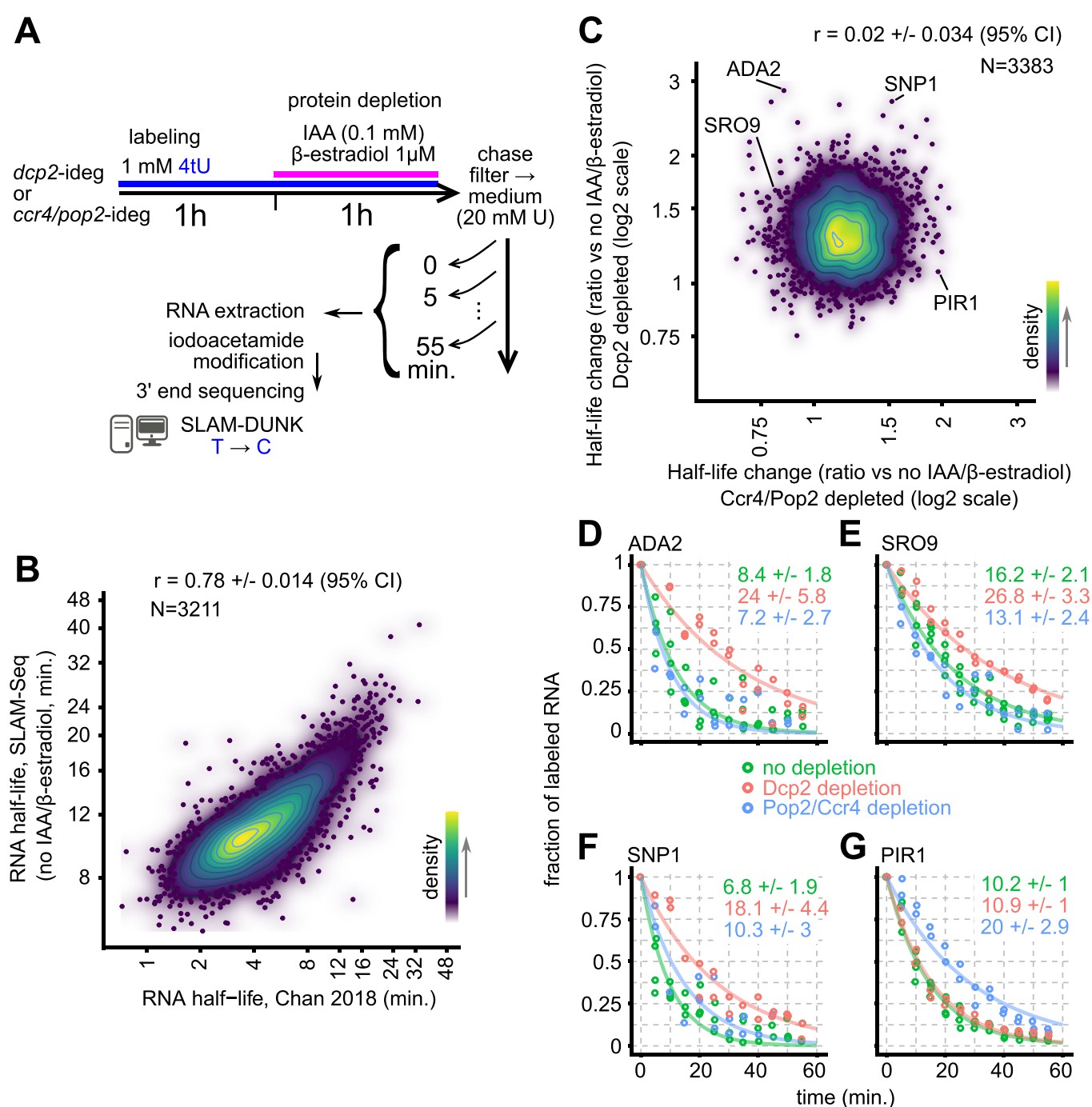

**Figure 4. mRNA half-life changes are mostly independent when decapping or deadenylation were inactivated.**

(A) Schematics of the SLAM-Seq experiment performed using *ccr4/pop2*-ideg and *dcp2*-ideg strains. (B) Correlation between the average half-life measured in our experiments and half-life reported previously by Chan et al, 2018. Pearson correlation coefficient, its confidence interval and number of measurements, N, are indicated. (C) Scatter plot showing the distribution of changes in half-life for transcripts when Ccr4 and Pop2 were depleted (x axis) or Dcp2 was depleted (y axis) in comparison with the average half-life of experiments in which degradation of the proteins was not induced. Individual transcripts are indicated and raw results for these transcripts, together with the estimated half-lives are indicated in panel (D) for ADA2, (E) for SRO9, (F) for SNP1, and (G) for PIR1. In each panel, the T to C conversion rate was normalized to 1 at time 0 and its change over time was plotted. The confidence interval for the estimates half-life values was computed using the "*confint*" function of the MASS R package (see Methods). Control, "no depletion" results are depicted in green, depletion of Ccr4 and Pop2 results in blue and depletion of Dcp2 results in orange.

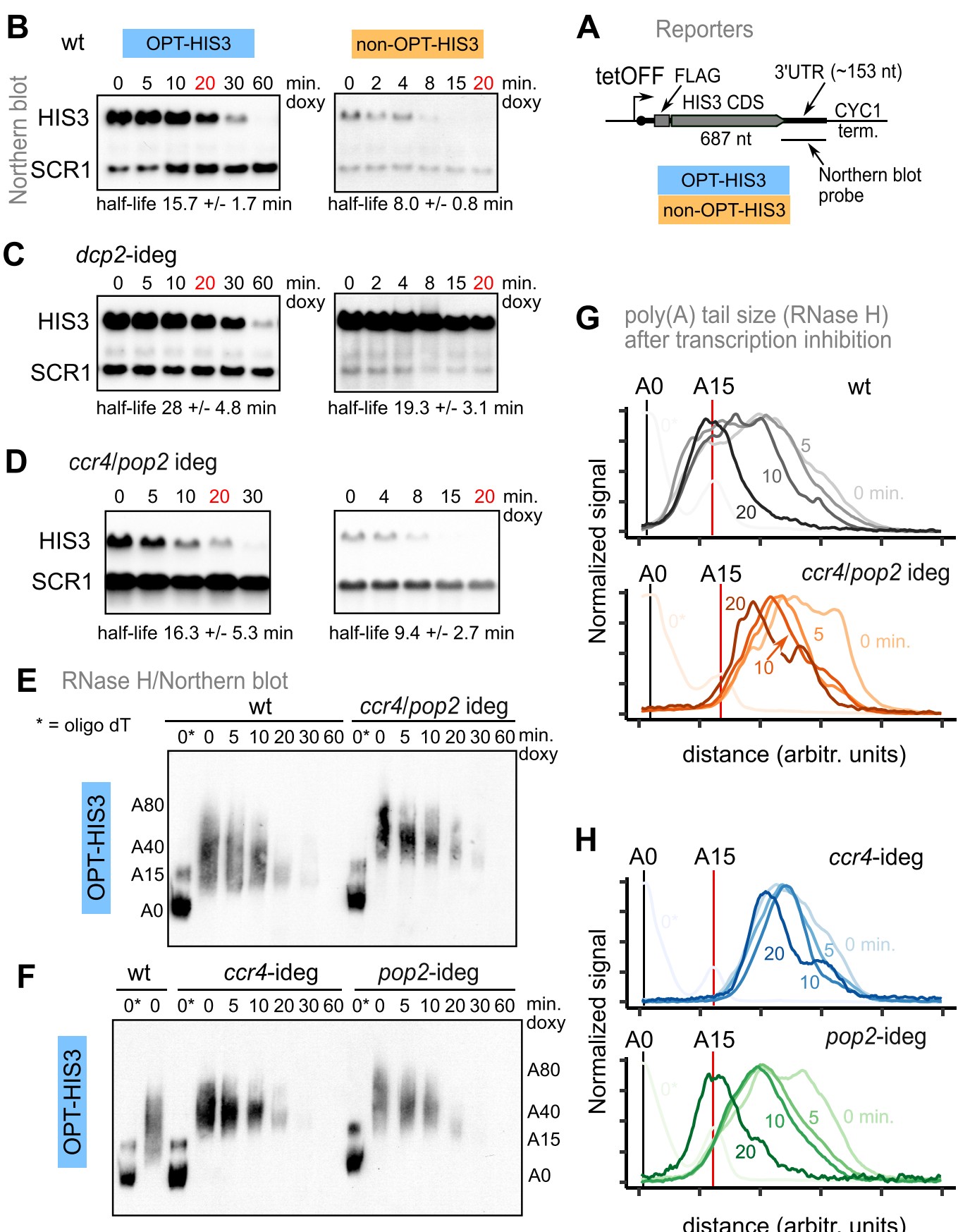

**Figure 5.  Deadenylation speed changes poly(A) size but does not affect half-life of reporter mRNAs.**

(A) Reporter mRNAs controlled by a tetOFF system were used for half-life estimates. (B) Northern blot visualization of degradation for reporter mRNAs in comparison with the stable SCR1 RNA, as control. The indicated half-life values correspond to quantitative experiments done by RT-qPCR (Fig. EV5). (C) Similar to (B) but following depletion of Dcp2. (D) Similar to (B) but following depletion of Ccr4 and Pop2 deadenylases. (E) Evolution of poly(A) tail length after transcription shut off with doxycycline in a wild-type strain (left side) or after depletion of Ccr4 and Pop2 through the inducible degron system (right side). A star indicates samples treated with oligo-dT prior to RNAse H digestion. Approximate sizes of the poly(A) tails are indicated. (F) Similar to (E), but for the individual depletion of either Ccr4 or Pop2. A wild-type RNA sample at time 0 is shown for comparison. (G) Poly(A) tail signal for the experiments presented in (E). The values were normalized to the same maximum signal and are not indicative of the amount of remaining RNA at different time points, only of the distribution. The partially digested signal corresponding to A15 was used to calibrate the plots. The horizontal axis corresponds to distance from the region of the A0 signal. (H) Similar to (G), for the poly(A) profiles corresponding to strains depleted for either Ccr4 or Pop2. Source data are available online for this figure.

To better understand the impact of blocking deadenylation on the poly(A) tail and degradation of the reporters, we tested both their steady-state and deadenylation dynamics by using RNAse H digestion and Northern blotting (Fig. 5E,F). Depletion of the deadenylases, either alone or in combination, drastically changed the steady-state distribution of poly(A) tails for both reporters. Whereas, in wild-type conditions, the steady-state distribution of poly(A) tails encompassed a broad range, from less than 10 to more than 80 A, a striking decrease in the signal corresponding to short poly(A) tails was observed following deadenylases depletion (Fig. 5E,G). While short poly(A) tails were not detectable for the reporter RNAs in cells depleted for Ccr4, they could be partially detected in the Pop2-depleted cells (Fig. 5F,H). These differences were comparable with those described for total RNA poly(A) tail dynamics after transcription shut-off, where CCR4 deletion led to a loss of deadenylated species shorter than 20–23 nucleotides (Webster et al, 2018).

## Discussion

Here we present several lines of evidence in favor of a deadenylation-independent model for the initiation of degradation of an important fraction of yeast mRNA. Our conclusions have implications for the understanding of how gene expression is regulated and suggest that deadenylation is not the main cause, but it rather accompanies RNA degradation. First, we found that the poly(A)-binding protein Pab1 is associated with unstable RNAs with longer than average poly(A) tails, many of them substrates of the NMD pathway (Fig. 1). Interestingly, and in agreement with previous studies, unstable mRNAs tend to have longer than average poly(A) tails not only in yeast, but also in other species, such as *C. elegans* (Lima et al, 2017), *A. thaliana* (Jia et al, 2022) or, to a lesser extent, human cells (Legnini et al, 2019). Second, we propose a simple kinetic model, with only two parameters, to explain steady-state poly(A) tail lengths as a function of RNA degradation rate (Fig. 2). In line with the previous investigation of reporter RNA degradation through modeling (Cao and Parker, 2001, 2003), we found that, to explain large-scale results, either deadenylation-independent decapping is prevalent, or deadenylation rate plays a minor role compared to decapping activation on oligoadenylated species. Third, to get a better and minimally perturbed view of the importance of deadenylation in RNA degradation, we constructed strains for rapid degradation of the major deadenylation factors Ccr4 and Pop2. Nanopore sequencing showed that, even though most RNAs carry longer poly(A) tails under these conditions, their relative levels were not affected. This was in contrast with the depletion of Dcp2, which clearly showed that the RNAs that had

initially the longer poly(A) tails and were most unstable, increased in their levels and showed a downshift in poly(A) tail length (Fig. 3). These results were further confirmed by measuring mRNA half-life in the same conditions using metabolic labeling (Fig. 4). Finally, the detailed investigation of the kinetics of RNA degradation for two reporter RNAs independent of NMD and having different degradation rates showed the lack of correlation between poly(A) status and rate of deadenylation and the degradation speed (Fig. 5).

Our results might seem at odds with the widely accepted idea that deadenylation precedes decapping in the degradation of RNA in eukaryotes. Still, recent results have already suggested that deadenylation is not linked with RNA degradation, at least during yeast meiosis (Wiener et al, 2021). Even the results of large-scale metabolic labeling and poly(A) tail length measurements in 3T3 mouse cells (Eisen et al, 2020) are compatible with a deadenylation-independent mechanism for RNA degradation. Individual highly unstable transcripts, such as Metrnl, showed a broad distribution of poly(A) tails at steady-state, while a very stable one, such as Eef2, displayed a more compact distribution, with shorter poly(A) tails on average. Individual cases were not reflected in a global correlation between poly(A) tail length and RNA stability, potentially because deadenylation and decapping have different ranges and dynamics in mammalian cells compared with yeast. Importantly, fitting a deadenylation-dependent model for RNA degradation involved a highly variable parameter, the speed of decapping for oligoadenylated RNA species. This speed varied by a factor of 1000 (Eisen et al, 2020), which is similar to the range of variation in the reported model for the deadenylation speed. Thus, the published model only fits experimental data if it includes a highly variable and limiting decapping speed. A strong correlation between measured deadenylation and decapping rates (Pearson correlation coefficient of 0.6, based on data presented in Table S1 of that article, 3140 transcripts) might indicate that unstable mRNA are particularly prone to deadenylation and that, once they reach a poly(A) threshold get decapped also rapidly. This result, based on the correlation of the parameters obtained by data fitting to a specific model, need to be critically assessed. For example, further examination of the reported parameters of the degradation kinetics of thousands of transcripts showed a strong positive correlation between deadenylation rate and RNA production rate (Pearson correlation coefficient 0.49) and also between decapping and RNA production (Pearson correlation coefficient of 0.51). These results would imply that the most highly produced RNAs have also the fastest deadenylation and decapping rates, a phenomenon that was not observed previously in mammalian cells (Schwanhäusser et al, 2011) and has no obvious biological meaning. This example illustrates why observed correlations between

parameters of a model could be misleading, unless they can be independently validated.

To circumvent the potential pitfalls of using correlation results for mechanistic conclusions we used both modeling and perturbation of decapping and deadenylation. Our initial modeling of RNA degradation kinetics identified the requirement for a strong variation in decapping speed for the deadenylation-dependent mechanism, to explain large-scale steady-state results (Fig. 2E). As demonstrated here, a deadenylation-independent mechanism could also lead to a similar distribution of poly(A) tails length and half-life (Fig. 2F). Previous simulations of RNA degradation based on the analysis of pulse-chase results for an unstable and stable RNA in yeast led to a similar conclusion: the stable RNA decapping should happen at much lower rates than the unstable one (Cao and Parker, 2001). Thus, even if degradation of RNA occurs only after deadenylation, a slow decapping rate could ensure the stability of an RNA, independent on its deadenylation rate. Our results bring to attention this important, and often overlooked element in RNA degradation. What mechanisms are involved in such a difference is unknown.

While our results and those of previous models are largely correlative, it was crucial to see what happens to RNA degradation when deadenylation or decapping is disrupted. A result that cannot be explained by a deadenylation-dependent model of RNA degradation is the lack of correlation between changes in poly(A) tail length and changes in RNA levels in deadenylation mutants, as shown here (Fig. 3D) and previously observed for *ccr4Δ* and *pan2Δ* (Tudek et al, 2021). These results were correlated with the lack of a stabilizing effect of depleting Ccr4, Pop2 or of both deadenylases on reporter RNAs, despite important changes in their poly(A) status (Fig. 5). In line with an independent effect of deadenylation and decapping on mRNA degradation is also the observation that mRNA stability changes were not correlated between conditions in which decapping or deadenylation were inhibited (Fig. 4).

Altogether, our large-scale purification, poly(A) tail and stability measurements, mRNA reporters and modeling results tend to show that deadenylation is not a cause of mRNA degradation, but rather a parallel process, that can occur at different rates for different transcripts. Further arguments for the uncoupling of deadenylation and RNA degradation can be found in published results from a seminal paper describing the importance of codon usage in RNA stability. In a chase experiment, the signal corresponding to oligo(dT) purified RNA for the highly stable PGK1 transcript was lost in 20 to 30 min, while the signal corresponding to non-polyadenylated RNA remained constant for over 60 min after transcription inhibition (Herrick et al, 1990). It is thus likely that the poly(A)+ fraction "degradation" rate was, in reality, a measure of progressive deadenylation with the loss of signal corresponding to a shift of RNA to shorter poly(A) tails. Such a discrepancy between the poly(A)+ and total RNA degradation rates has also been observed more recently on a large scale (Presnyak et al, 2015) and could thus be explained, at least in part, by the fact that measuring signal loss in the poly(A)+ fraction corresponds to measuring deadenylation and RNA degradation at the same time.

Technical improvements in assessing RNA degradation on a large scale are likely to overcome some of the limitations of our study. For example, when working with total RNA, we could not account for variations in the fraction of each mRNA in different intracelullar compartments. Subcellular fractionation coupled with metabolic labeling can overcome this limitation, as recently demonstrated by experiments on mouse and human cell lines (Ietswaart et al, 2024). Interestingly, this work identified a slight trend towards relatively longer poly(A) tails for unstable cytoplasmic mRNAs compared with stable ones, which is consistent with our results in yeast. In addition to cellular compartments, two other factors may influence our understanding of mRNA degradation. One is the ratio of full-length mRNA to mRNA degradation intermediates, and the other is the relative solubility of different mRNA forms, as recently demonstrated in yeast (Allen et al, 2023). Finally, when cells are affected by depletion of key mRNA degradation factors, such as deadenylases or the decapping enzyme, changes in the transcriptome can lead to a redistribution of RNA-binding proteins that may diferentially affect the stability or metabolism of transcripts. This type of effect has been demonstrated, for example, by showing that perturbation of the nuclear exosome leads to accumulation of RNA that, through binding of transcription termination factors indirectly affects transcription (Villa et al, 2020). Therefore, the developement of more sophisticated systems that track a range of reporter mRNA molecules in minimally perturbed cells is needed to further analyze RNA degradation mechanisms.

Our results clearly demonstrate that a re-evaluation of the deadenylation-dependent and deadenylation-independent models of mRNA degradation would help to better understand the complex relationship between the poly(A) tail, translation and RNA stability. The results presented here should encourage the use of alternative models for mRNA degradation and stimulate further research to characterize the underlying molecular mechanisms.

# Methods

**Reagents and tools table**

| Reagent/Resource | Reference or Source | Identifier or Catalog Number |
|---|---|---|
| **Experimental Models** | | |
| NEB 10-beta competent *E. coli* | New England Biolabs | C3019H |
| LMA2154 (BY4741) (*S. cerevisiae*) *MATa ura3Δ0 his3Δ1 leu2Δ0 met15Δ0* | (Brachmann et al, 1998) | N/A |
| LMA5395 *MAT a tao3*(E1493Q) *rme1* insert A-308 *trp1*(D87Stop) (*trp1-1*) | (This study and Bushkin et al, 2019) | N/A |
| LMA5396 *MAT a tao3*(E1493Q) *rme1* insert A-308 *trp1*(D87Stop) (*trp1-1*) | (This study and Bushkin et al, 2019) | N/A |
| LMA5419 *Mat a tao3*(E1493Q) *rme1* insert A-308 *trp1-1::TRP1-prURA3-Z3EVtf-tURA3-Z3EV-TIR1* | (This study and Bushkin et al, 2019) | N/A |
| LMA5575 - LMA5419 + *dcp2*-polyG-3FLAG-AID::KANMX6 | This study | N/A |
| LMA5593 - LMA5419 + AID-3FLAG-*ccr4*::KANMX6 | This study | N/A |
| LMA5597 LMA5419 + *pop2*-polyG-3FLAG-AID::KANMX6 | This study | N/A |

| Reagent/Resource | Reference or Source | Identifier or Catalog Number |
|---|---|---|
| LMA5601 - LMA5419 + AID-3FLAG-*ccr4 pop2*-AID-polyG-3FLAG-AID::KANMX6 | This study | N/A |
| LMA2194 - BY4741 + *nam7(upf1)*-TAP::HIS3MX6 | (Ghaemmaghami et al, 2003) | N/A |
| LMA2195 - BY4741 + *rpl16a*-TAP::HIS3MX6 | (Ghaemmaghami et al, 2003) | N/A |
| LMA4987 - BY4741 + *pab1*-TAP::HIS3MX6 | (Ghaemmaghami et al, 2003) | N/A |
| LMA4985 - BY4741 + *lsm1*-TAP::HIS3MX6 | (Ghaemmaghami et al, 2003) | N/A |
| LMA4986 - BY4741 + *lsm7*-TAP::HIS3MX6 | (Ghaemmaghami et al, 2003) | N/A |
| LMA2356 - BY4741 + *pat1*-TAP::HIS3MX6 | (Ghaemmaghami et al, 2003) | N/A |
| LMA5243 - BY4741 + *dhh1*-TAP::HIS3MX6 | (Ghaemmaghami et al, 2003) | N/A |
| LMA1667 - BY4741 *nam7(upf1)*Δ::KANMX6 | (Giaever et al, 2002) | N/A |
| LMA5018 - BY4741 *pab1*-HA::KANMX6 | This study | N/A |
| LMA5028 - BY4741 *pab1*-HA::KANMX6 *nam7(upf1)*-TAP::HIS3MX6 | This study | N/A |
| **Recombinant DNA** | | |
| pCM189-NFLAG-HIS3-10 p1638 (TETO7-NFLAG-HIS3-10) (URA3) | This study | N/A |
| pCM189-NFLAG-HIS3-100 p1640 (TETO7-NFLAG-HIS3-100) (URA3) | This study | N/A |
| pFA6a-polyG-3FLAG-miniAID-opt-KANMX6 - p1451 | (Dehecq et al, 2018) | N/A |
| pRS304_pURA3-Z3EVtf-tURA3 - pl. 1603 | This study | N/A |
| pURA3-Z3EVtf-tURA3-Z3EV-TIR1 - pl. 1692 | This study | N/A |
| pRS315 (LEU2) CEN | (Sikorski and Hieter, 1989) | N/A |
| pRS315-CCR4 (LEU2) CEN | This study | N/A |
| pRS315-CCR4(E556A) (LEU2) CEN | This study | N/A |
| pRS315-POP2 (LEU2) CEN | This study | N/A |
| pRS315-POP2(D310A) (LEU2) CEN | This study | N/A |
| **Antibodies** | | |
| Mouse anti-FLAG M2 - HRP | Sigma-Aldrich | A8592 |
| Rabbit Peroxidase Anti-Peroxidase soluble complex | Sigma-Aldrich | P1291 |
| Rat anti-HA - HRP | Roche | 12013819001 |
| Sheep anti-digoxigenin-POD (poly), Fab fragments | Roche | 11633716001 |
| **Oligonucleotides and other sequence-based reagents** | | |
| PCR primers | This study | Dataset EV5 |

| Reagent/Resource | Reference or Source | Identifier or Catalog Number |
|---|---|---|
| **Chemicals, Enzymes and other reagents** | | |
| Phusion high-fidelity DNA polymerase | ThermoFisher | F530L |
| Q5 high-fidelity DNA polymerase | New England Biolabs | M0491 |
| Platinum SuperFi II DNA Polymerase | ThermoFisher | 12361010 |
| DNA polymerase I, Large (Klenow) Fragment | New England Biolabs | M0210 |
| Superscript III reverse transcriptase | ThermoFisher | 18080044 |
| RNase H | New England Biolabs | M0297 |
| T5 exonuclease | New England Biolabs | M0663 |
| Terminal transferase | New England Biolabs | M0315 |
| Dynabeads M-270 epoxy | ThermoFisher | 14302D |
| Rabbit IgG | Sigma-Aldrich | I5006 |
| DIG RNA labelling kit (SP6/T7) | Roche | 11175025910 |
| Digoxigenin-11-dUTP | Roche | 11093088910 |
| pBR322 | New England Biolabs | N3033 |
| Indole-3-acetic acid (IAA) | Sigma-Aldrich | I2886 |
| Beta-estradiol | Sigma-Aldrich | E2758 |
| Doxycycline | Sigma-Aldrich | D3447 |
| 4-Thiouracil | Sigma-Aldrich | 440736 |
| *Bam*HI-HF | New England Biolabs | R3136 |
| *Bsu*36I | New England Biolabs | R0524 |
| *Not*I-HF | New England Biolabs | R3189 |
| *Sst*II | ThermoFisher | 45230-018 |
| *Sma*I | New England Biolabs | R0141 |
| *Xho*I | New England Biolabs | R0146 |
| Lysyl endopeptidase (Lys-C) | Fujifilm Wako chemicals | 125-05063 |
| Trypsin gold, mass spectrometry grade | Promega | V5280 |
| Blocking reagent (nucleic acids detection) | Roche | 11096176001 |
| Oligo d(T)$_{25}$ magnetic beads | New England Biolabs | S1419S |
| TruSeq Stranded mRNA sequencing kit | Illumina | 20020594 |
| Quantseq 3' mRNA Seq V2 with UDI | Lexogen | 194.96 |
| Ribo-Zero Gold rRNA (Yeast) | Illumina | MRZY1324 |
| RiboMinus™ Transcriptome Isolation Kit, yeast | ThermoFisher | K155003 |
| Ribonucleoside vanadyl complex | New England Biolabs | S1402 |
| RNasin ribonuclease inhibitor | Promega | N2511 |
| SsoAdvanced Universal SYBR Green Supermix | Bio-Rad | 1725271 |
| DNase TURBO | ThermoFisher | AM2238 |
| Micrococcal nuclease | New England Biolabs | M02475 |

| Reagent/Resource | Reference or Source | Identifier or Catalog Number |
|---|---|---|
| cOmplete protease inhibitor mix (no EDTA) | Roche | 11873580001 |
| Direct RNA Sequencing Kit | Oxford Nanopore | SQK-RNA002 |
| Clarity Western ECL substrate | Bio-Rad | 1705061 |
| **Software** | | |
| ImageJ (1.54) | National Institutes of Health (NIH) https://imagej.net/ij/ | N/A |
| Image Lab Software (6.1) | Bio-Rad | 12012931 |
| CFX Maestro Software (qPCR) | Bio-Rad | 12013758 |
| Inkscape (1.3.2) | https://www.inkscape.org | N/A |
| LibreOffice (7.6) | https://www.libreoffice.org/ | N/A |
| STAR (2.7) | (Dobin et al, 2013) | N/A |
| Integrated Genome Viewer (IGV, 2.11) | (Robinson et al, 2011) | N/A |
| SLAM-DUNK | (Neumann et al, 2019) | N/A |
| R (3.8-4.4) | (R Core Team, 2024) https://www.R-project.org/ | N/A |
| RStudio | https://posit.co/products/open-source/rstudio/ | N/A |
| Guppy | Oxford Nanopore | N/A |
| Minimap2 (2.1) | (Li, 2018) | N/A |
| Samtools (1.9) | (Danecek et al, 2021) | N/A |
| Nanopolish (0.13.2) | (Workman et al, 2019) | N/A |
| MaxQuant (2.0.30) | (Cox and Mann, 2008) | N/A |
| Perseus (1.6.15) | (Tyanova et al, 2016) | N/A |
| Tellurium | (Choi et al, 2018) | N/A |
| wxMaxima (20.12.1) | https://github.com/wxMaxima-developers/wxmaxima/ | N/A |
| ApE (2.0.61) | (Davis and Jorgensen, 2022) | N/A |
| **Other** | | |
| NuPAGE 4–12% Bis-Tris electrophoresis gel | ThermoFisher | NP0323BOX |
| Trans-Blot Turbo transfer system | Bio-Rad | 1704150 |
| Amersham Hybond-N+ | Cytiva Life Sciences | RPN303B |
| Supported nitrocellulose membrane | Bio-Rad | 1620094 |
| Gel and PCR clean-up | Macherey-Nagel | 740609.250 |
| NextSeq 500 | Illumina | N/A |
| LTQ Orbitrap Velos mass spectrometer (nanoLC Ultimate 3000) | ThermoFisher | N/A |
| MinION sequencing device | Oxford Nanopore | N/A |
| ChemiDoc XRS+ imaging device | Bio-Rad | N/A |

## Yeast and bacterial strains

*Saccharomyces cerevisiae* strains were derived from BY4741 (Mat a) and BY4742 (Mat α) strains. Strain GBy68 (Bushkin et al, 2019) was kindly provided by G. Bushkin, Whitehead Institute for Biomedical Research, Cambridge, MA, USA. An alpha haploid spore derived from this strain, with the *rme1-Δ*, ins-308A and the *tao3*(E1493Q) alleles was selected and crossed with a *trp1*-1 (i.e., trp1(D87-stop)) derivative of BY4741. After sporulation of the resulting diploid cell, spores with the following genotype Mat a, *LYS2*, *met15Δ0*, *ura3Δ0*, *leu2Δ0*, *his3Δ1*, *trp1*-1, *RME1-Δ*, ins-308A, *TAO3*(E1493Q) or Mat alpha, *lys2Δ0*, *ura3Δ0*, *leu2Δ0*, *his3Δ1*, *trp1*-1, *RME1-Δ*, ins-308A, *TAO3*(E1493Q) were selected to create strains LMA5395 and LMA5396 respectively. LMA5395 was transformed with plasmid p1692 linearized by *Bsu*36I digestion. A tryptophan prototroph transformant was selected to create strain LMA5419.

*E. coli* strain NEB 10-beta (NEB Cat# C3019) was used for construction of plasmids by ligation-less Gibson assembly (Fu et al, 2014) and their multiplication. All the plasmid inserts were verified by sequencing.

C-terminal TAP-tagged strains originated from the collection of systematically built strains (Ghaemmaghami et al, 2003). Deletion strains were part of the systematic yeast gene deletion collection (Giaever et al, 2002) distributed by EuroScarf (http://www.euroscarf.de) or were built by transformation of BY4741 strain with a cassette containing a selection marker cassette flanked by long recombination arms located upstream and downstream the open reading frame of the gene. Deletions were tested by PCR amplification of the modified locus.

To create auxin-inducible degron strains, the indicated genes were C-terminally tagged with a polyG-3Flag-miniAID-kanMX6 cassette amplified by PCR using plasmid p1451 as a template (Dehecq et al, 2018) in LMA5396. The resulting strains were then crossed with LMA5419 and sporulated to recover strains of the desired genotype. The only exception was the CCR4-degron strain for which the AID-tag amplified from plasmid p1602 has been inserted at the N-terminus of the protein by CRISPR/CAS9 engineering as described (Mans et al, 2015).

## Media and growth conditions

Yeast cells were grown in YPD (20 g.L$^{-1}$ glucose, 10 g.L$^{-1}$ yeast extract, 20 g.L$^{-1}$ bacto-peptone, 20 g.L$^{-1}$ bactoagar for plates only) and in synthetic media without uracil or leucine to select transformants and maintain plasmids with the URA3 or LEU2 marker. All yeast strains were freshly thawed from frozen stocks and grown at 30 °C. Bacterial strains were grown in LB media, supplemented with antibiotics when necessary, at 37 °C.

## Plasmids construction

Sequences manipulation were performed using ApE (Davis and Jorgensen, 2022). Plasmids for the His3 codon-changed reporters: coding sequences for His3 versions, «10%» and «100%» were recovered from published data (Radhakrishnan et al, 2016) and are also presented in Appendix Table S1. The DNA fragments were synthesized by Twist Bioscience (San Francisco, CA, USA). The coding sequences were amplified with LA070 and LA071

before Gibson assembly reaction. The final PCR product was cloned into the *BamH*I and *Not*I sites of pCM189 (Garí et al, 1997).

Plasmids for the degron system: for the p1603 plasmid, the Z3EV artificial transcription factor was PCR amplified from plasmid pFS461 (Ohira et al, 2017) using oligonucleotides FF3765 and FF3766 and the PCR product was used to transform a URA3 derivative of BY4741. 5-FOA resistant cells were selected and checked for correct replacement of the URA3 coding sequence by the Z3EV artificial transcription factor coding sequence. The *ura3::Z3EV* allele obtained was amplified from yeast genomic DNA using oligonucleotides FF293 and FF3746. The final PCR product was digested with *Sst*II and cloned into the *Sst*II and *Sma*I sites of pRS304.

For the p1692 plasmid, the Z3EV promoter was PCR amplified from plasmid pFS478 (Ohira et al, 2017) using oligonucleotides FF3740 and FF3872. The obtained PCR product was digested with *Sal*I and *BamH*I. The *O. sativa* TIR1 ORF followed by the Nrd1 terminator was PCR amplified using oligonucleotides FF3718 and FF3719 from the genomic DNA of a *nrd1*-AID strain kindly provided by D. Challal (Domenico Libri laboratory, IJM, Paris, France). The PCR product was digested with *BamH*I and *Xho*I. Both PCR products were cloned together into the *Xho*I site of plasmid p1603.

CCR4 and POP2 sequences containing the coding regions and flanking upstream (503 nt. for CCR4 and 500 nt for POP2) and downstream (560 nt for CCR4 and 527 nt. for POP2) sequences were cloned in pRS315 (Sikorski and Hieter, 1989) digested with *Not*I and *BamH*I. Catalytically affected versions of CCR4 (E556A, GAA to GCA) and POP2 (D310A, GAT to GCT) were obtained using amplification of DNA with oligonucleotides with the desired changes (CS1948 to CS1959, Dataset EV5).

## Cell culture, RNA extraction, and degradation kinetics

For RT-qPCR and RNA sequencing, cells were first grown in YPD to log phase and collected. Total RNA was extracted using the hot phenol extraction method and precipitated using ammonium acetate and ethanol.

For the tetOFF promoter inhibition and steady state analysis, cells expressing the appropriate plasmids were grown at 25 °C in synthetic media without uracil. For the analysis of degron mutants, when cells reached a OD600 of 0.4, β-estradiol for Tir1 expression at a final concentration of 1 μM and IAA (auxin) at a final concentration of 100 μM for protein depletion were added directly to the media. Several time points were collected to verify protein depletion by western blot with an anti-FLAG HRP antibody. After 1 h, cells were harvested for steady state analysis. For transcriptional repression, doxyxycline was added at a final concentration of 10 μg/mL. Cells were collected at the time points indicated in the Northern Blot, generally, 0, 2, 4, 8, 10, 20, 30 min for the non-OPT-HIS3 RNA and 0, 5, 10, 20, 30, 60 for the more stable, OPT-HIS3 RNA.

For translation arrest, cycloheximide was added at a final concentration of 5 to 50 μg/mL together with the doxycycline.

## Northern blots

RNAs were separated on 1.5% agarose gels, transferred on nylon membrane (Hybond N+, Amersham, GE) that was UV cross-linked at 0.120 Joules and probed with DIG-labeled RNA or DNA probes. Digoxigenin-containing RNA probes were generated by in vitro transcription with T7 polymerase using the DIG RNA Labelling kit (SP6/T7) from Roche (cat. No 11175025910). The oligonucleotide AJ529 (T7 promoter) pre-annealed with a target-specific oligonucleotide composed of the T7 promoter reverse complement fused to a template sequence, LA94 for His3-tCyc and LA95 for SCR1 were used as a template. Specific DNA probes for the NON-OPT and OPT HIS3 RNA were generated using MFR917 and LA084. After PCR amplification and purification, a single-stranded amplification was done using MFR918 for the OPT and the non-OPT version of the probes.

For poly(A) tail visualization, RNAs were cleaved using RNASE H (New England Biolabs, Ipswich, MA, USA) with LA121 as a specific primer for HIS3-tCYC1. Oligo-dT were added to the reaction to remove poly(A) tails for the «A0» reference position. RNAs were then purified with acid phenol-chloroform extraction. Cleaved RNAs were separated on a 6% acrylamide gel, transferred to a nylon membrane (Hybond N+, Amersham) and UV cross-linked. For the DNA ladder, a denatured PBR322 DNA digested with *Msp*I was used. Prior to the assay, digoxigenin-11-ddUTP was added to the 3′ end of the digested fragments using terminal transferase enzyme (New England Biolabs, Ipswich, MA, USA).

## RNA half-life measurements (RT-qPCR)

A mix of reverse qPCR oligonucleotides, LA126 (Act1), LA102 (His3-tCyc) and random hexamers were used for the reverse transcription. Serial dilutions of cDNA were quantified by TaqMan probe-based quantitative PCR. The amplification was done with the Bio-Rad CFX96 machine and the corresponding software (CFX Maestro Software), with step 1 (95 °C for 30 s) and step 2 (40 cycles of 95 °C for 15 s and 60 °C for 30 s). Probes were LA122-5′-Cy5/TAO/3′IBRQ for His3-tCyc and LA125-5′-HEX/ZEN/3′IBFQ for Act1. Primers were LA102 and LA123 for HIS3, LA124 and LA126 for ACT1.

Half-life estimates from RT-qPCR results were done using nonlinear regression on values that were calculated as a fraction from RNA levels at time 0, using the exponential decay function $e^{-(t-l)*k}$. In this fit, "t" is time after doxycyclin addition, "l" is a lag period (Baudrimont et al, 2017), estimated from the results obtained with the wild-type condition, and "k" is the decay constant. Estimates of half-life were obtained by the formula $(\ln(2)/k)+l$, that takes into account the lag period. Initial estimates of this lag, based on both non-OPT HIS3 and OPT-HIS3 reporters, led to a value of 1.2 min. that was used throughout to obtain estimates. The *confint* function from the "MASS" R package was used to obtain confidence estimates for "k" and associated half-life values at a 0.95 level of confidence.

## RNA half-life estimates (SLAM-seq)

Cells were grown in synthetic medium containing low uracil (10 mg/L, Half-URA) for two doubling times (OD ~ 0.25). To pulse label mRNA, we added 4-thiouracil to the culture media to a final concentration of 1 mM for 60 min. To deplete the targeted proteins, both β-estradiol (1 μM) and IAA (100 μM) were added to the media (or ethanol for the undepleted control), in the presence of 4-thiouracil, and the cells were incubated for further 60 min. To chase, the cells washed in Half-Ura medium with 1 μM β-estradiol and 100 μM IAA and immediately resuspended in synthetic

medium containing excess Uracil (20 mM) and no 4-thiouracil and either β-estradiol and IAA or etanol only for the undepleted control. Samples were collected by filtration every 5 min and frozen in liquid nitrogen. Next, the frozen cell pellets were resuspended in TES buffer (10 mM Tris HCl pH 7.5, 10 mM EDTA, 0.5% SDS) and the RNA was extracted by hot phenol/chloroform treatment and precipitated by isopropanol in the presence of sodium acetate. RNA alkylation was done with 10 mM iodoacetamide in 30% DMSO and 50 mM NaPO$_4$ (pH 8). Library preparation was performed using the Quantseq 3′ mRNA Seq V2 with UDI library preparation kit (Lexogen, Austria). The experiment was performed in two independent replicates. Reads were mapped as described in (Alalam et al, 2022) and the quantification of labeled transcripts was performed using the SLAM DUNK package (Neumann et al, 2019).

The output of "SLAM DUNK" was filtered to retain only transcripts with a minimum of 20 reads and conversion ratios inferior to 0.0001 were also removed. Half-life of transcripts was estimated using the exponential decay equation $A_t = A_0 * e^{-kt}$, where $A_t$ is the relative fraction of 4-thiouracil in the RNA at time (t), and $A_0$ the fraction of 4-thiouracil at time 0 before chase. "t" is time, in minutes, and "k" is the first-order decay constant for de RNA degradation. Half-life was computed as ln(2)/k, and corresponds to an $A_t/A_0$ ratio of 0.5. No correction for the dilution of labeled molecules by cell growth and division during the chase period was performed. A pseudo-determination coefficient for the nonlinear regression was used as an estimate of the goodness of fit for each set of results. Only situations with a pseudo-R$^2$ value (*rsquare* function from the *modelr* R package) superior to 0.8 were considered for further calculations. For each decay constant estimation, a 95% confidence interval was computed using the *confint* function of the *MASS* R package. The average of log2(half-life) values for the four experiments in which no IAA or β-estradiol were added to the cultures served as the base for comparison with the results obtained following Dcp2 or Ccr4/Pop2 depletion.

## Affinity purification for RNA sequencing, immunoblot, and mass-spectrometry

TAP-tagged proteins were purified using a one-step purification method (details in Namane and Saveanu, 2022). Frozen cell pellets of 4 L culture were resuspended in lysis buffer (20 mM HEPES K pH 7.4, 100 mM KOAc, 0.5% Triton X100, 5 mM MgCl$_2$, Protease inhibitor, 1X Vanadyl-Ribonucleoside Complex, 40 units ml$^{-1}$ RNasin) and lysed using a MagNA Lyser (two passages of 90 s at 4000 rpm). The lysate was cleared at 4 °C for 20 min at 14,000 rpm. Magnetic beads (Dynabeads M-270 epoxy) coupled to IgG were added to the protein extract and incubated for 1.5 h at 4 °C (Oeffinger et al, 2007). Beads were magnetically separated and extensively washed five times with a washing buffer (20 mM HEPES K pH 7.4, 100 mM KOAc 0.5% Triton X100, 5 mM MgCl$_2$). After washing, proteins were eluted by incubation in denaturing buffer (SDS 2% in Tris-EDTA pH 7.5) 10 min at 65 °C. After collection, RNA were extracted by hot acid phenol/chloroform method, precipitated with ammonium acetate and ethanol. Samples were treated by RiboZero (Illumina) or Ribominus (Thermo Scientific) rRNA removal reagents and libraries were prepared using TruSeq Stranded mRNA kit (Illumina). We followed the protocol except that we started directly at the fragmentation step after the rRNA

removal step and we adjusted the number of cycles required to amplify libraries. For RNAse treatment, 1 µl of micrococcal nuclease (Biolabs M02475, 2 × 10$^6$ U/ml) was added to 200 µl buffer containing 1 mM CaCl$_2$ and incubated for 10 min 37 °C. An RNA aliquot was harvested prior to TAP purification for the analysis of total RNAs (input) and was treated the same way as the purified fractions except for the purification.

## Illumina RNA sequencing results analysis

Reads were aligned along the 16 chromosomes and mitochondrial sequence of *S. cerevisiae* S288C genome. For the mapping, we used the STAR program (version 2.7) with default parameters except for the following preferences: alignIntronMax 1500, alignMatesGap-Max 1500, alignSJoverhangMin 25 (Dobin et al, 2013). For the exon annotation we used the GTF file for *Saccharomyces cerevisiae* (R64-1-1.104) from ENSEMBL. Read counts were obtained using the featureCounts function of the Subread package, with an annotation file that contains non-coding RNA coordinates in addition to yeast gene transcripts (Malabat et al, 2015). The analyzed data was visualized using the Integrated Genome Viewer, IGV (Robinson et al, 2011) and further data processing was done using R (R Core Team, 2022). It consisted in filtering out features with less than 10 reads, followed by a normalization by the total number of reads for each sample and averaging of the log2 transformed results for the three replicated experiments for each condition. Input normalized counts were substracted from purification counts for each condition. Finally, the results were adjusted by substracting the median of the log2 enrichment values for each quantified feature. For the comparison with RT-qPCR results, raw enrichment values were calculated by substracting, for each replicate, the fraction of counts in input samples from the fraction of counts in each purified sample (log2 transformed values).

## Reverse transcription and quantitative PCR

The extracted RNA samples were treated with DNase I (Ambion TURBO DNA-free kit) before reverse-transcription (RT) with the Superscript II or III (Invitrogen). The purified and input samples were used for reverse transcription with transcript-specific primers. A mix of reverse qPCR oligonucleotides, CS888 (RPL28-premature), CS889 (RPL28) and random hexamers were used for the RT. Serial dilutions of cDNA were quantified by qPCR, CS887-CS888 (RPL28-premature), and CS889-CS946 (RPL28). The amplification was done in a Bio-Rad CFX96, with step 1 (95 °C for 3 min) and step 2 (40 cycles of 95 °C for 10 s and 60 °C for 30 s).

## Protein extracts and immunoblots

Total protein extracts were prepared from 5 ml of exponential culture cells using an alkaline treatment. Cells were incubated in 200 µL of 0.1 M NaOH for 5 min at room temperature, collected by 3 min centrifugation and resuspended in 50 µL of SDS sample buffer containing DTT (0.1 M). Proteins were denatured for 3 min at 95 °C, and cellular debris were pelleted by centrifugation. 10 µL of supernatant or diluted supernatant (for quantification scale) were loaded on acrylamide NuPAGE Novex 4–12% Bis-Tris gels (Life technologies). Transfer to a nitrocellulose membrane was done with a semi-dry fast

system (Biorad trans-blot) with discontinuous buffer (BioRad technote 2134). Proteins were detected by hybridization with anti-FLAG-HRP, for the detection of the FLAG tag (Sigma-Aldrich, A8592, clone M2, monoclonal, RRID:AB_439702), PAP (Sigma-Aldrich P1291, peroxidase anti-peroxidase soluble complex antibody, RRID:AB_1079562), for the detection of the protein A fragment of the TAP tag or anti-HA peroxidase 1/500 (clone 3F10, Roche Cat# 12013819001, RRID:AB_390917).

## Mass spectrometry acquisition and data analysis

Protein samples were treated with Endoprotease Lys-C (Fujifilm Wako chemicals, Osaka, Japan) and Trypsin (Trypsin Gold Mass Spec Grade, Promega). Peptide samples were desalted using OMIX C18 pipette tips (Agilent Technologies). The peptides mixtures were analyzed by nano-LC-MS/MS using an Ultimate 3000 system (Thermo Fisher Scientific) coupled to an LTQ-Orbitrap Velos mass spectrometer. Peptides were desalted on-line using a trap column (C18 Pepmap100, $5\,\mu m$, $300\,\mu m$Å~$5\,mm$ (Thermo Scientific) and then separated using $120\,min$ RP gradient (5–45% acetonitrile/0.1% formic acid) on an Acclaim PepMap100 analytical column (C18, $3\,\mu m$, 100, $75\,\mu m$ id × $150\,mm$ (Thermo Scientific) with a flow rate of $0.340\,\mu L.min^{-1}$. The mass spectrometer was operated in standard data-dependent acquisition mode controlled by Xcalibur 2.2. The instrument was operated with a cycle of one MS (in the Orbitrap) acquired at a resolution of 60,000 at $m/z$ 400, with the top 20 most abundant multiply-charged (2+ and higher) ions subjected to CID fragmentation in the linear ion trap. An FTMS target value of $1e^6$ and an ion trap MSn target value of 10,000 were used. Dynamic exclusion was enabled with a repeat duration of $30\,s$ with an exclusion list of 500 and exclusion duration of $60\,s$. Lock mass of 445.12002 was enabled for all experiments. The results were analyzed with MaxQuant (Cox and Mann, 2008) and processed with Perseus (Tyanova et al, 2016) and R, as previously described (Dehecq et al, 2018). Enrichment calculations were based on a standard dataset of protein abundance in yeast (Ho et al, 2018). At least three independent experiments were performed for each purified protein.

## Nanopore sequencing and analysis

Degradation of target proteins was induced for $1\,h$ in exponential growth cultures in rich (YPD) medium, with ß-estradiol ($1\,\mu M$ final) and IAA ($100\,\mu M$ final). Protein depletion was verified by immunoblot against the FLAG tag. Total RNA was extracted using a hot phenol protocol. RNA libraries were prepared from 250 to $500\,ng$ of oligo-(dT)25-enriched mRNA with a Direct RNA Sequencing Kit (catalog no. SQK-RNA002, Oxford Nanopore Technologies) according to the manufacturer's instructions. Sequencing was performed using R9.4 flow cells on a MinION device (ONT). Raw data were basecalled using Guppy (ONT). Reads were mapped to the SC288 reference genome Saccer3.fa using minimap2.1 with options "-k 14 -ax map-ont –secondary = no" and processed with samtools 1.9 (samtools view -b -o) (Danecek et al, 2021). The poly(A) tail lengths for each read were estimated using the Nanopolish 0.13.2 polya function (Workman et al, 2019). In subsequent analyses, only length estimates with the QC tag that were reported by Nanopolish as "PASS" were considered. We also retained only features with at least 20 assigned reads in two independent replicates. Statistics of

obtained results, poly(A) tail length and counts, were computed with R.

## Simulation of RNA degradation and deadenylation

To be able to estimate how global poly(A) tail would change with RNA stability, we used two complementary approaches. In one, we obtained, using Maxima (Maxima, 2022), an ordinary differential equation solving system, the equations describing the evolution over time of three species of RNA of different poly(A) length (long, medium, and short) under two kinetic models. The deadenylation-dependent, or serial, model described by the following expressions:

$$\text{ser1:}'\ \text{diff}(PAl(t), t) = T - kA * PAl(t);$$

$$\text{ser2:}'\ \text{diff}(PAm(t), t) = kA * PAl(t) - kA * PAm(t);$$

$$\text{ser3:}'\ \text{diff}(PAs(t), t) = kA * PAm(t) - kD * PAs(t).$$

Here, "T" is constant and represents all the events preceding RNA deadenylation and degradation (synthesis, splicing, nuclear export), "kA" is a deadenylation constant (pseudo-first order process) that dictates the rate of transformation of long to medium and to short poly(A) mRNAs and "kD" is a degradation constant, corresponding to the transformation of the RNA to decay products. "t" represents time. Solving the system involves the following Maxima commands:

$$\text{atvalue}(PAl(t), t = 0, 0); \text{atvalue}(PAm(t),$$
$$t = 0, 0); \text{atvalue}(PAs(t), t = 0, 0);$$

$$\text{desolve}([\text{ser1}, \text{ser2}, \text{ser3}], [PAl(t), PAm(t), PAs(t)]);$$

For the deadenylation-independent, or parallel, model, the equations included an additional step of degradation for both PAl and PAm species:

$$\text{par1:}'\ \text{diff}(PX1(t), t) = T - kA * PAl(t) - kD * PAl(t);$$

$$\text{par2:}'\ \text{diff}(PX2(t), t) = kA * PAl(t) - kA * PAm(t) - kD * PAm(t);$$

$$\text{par3:}'\ \text{diff}(PAs(t), t) = kA * PAm(t) - kD * PAs(t);$$

$$\text{atvalue}(PAl(t), t = 0, 0); \text{atvalue}(PAm(t),$$
$$t = 0, 0); \text{atvalue}(PAs(t), t = 0, 0);$$

$$\text{desolve}([\text{par1}, \text{par2}, \text{par3}], [PAl(t), PAm(t), PAs(t)]);$$

For steady-state conditions, with "t" reaching very high values, the obtained results indicated that PAl and PAm accumulation would be only dependent on deadenylation, $k_A$, rates in the deadenylation-dependent model. PAs accumulation at steady state would be only dependent on $k_D$, as expected. For the deadenylation-independent model, the relative levels of the different species of RNA at steady state depend on both $k_A$ and $k_D$.

While the obtained equations could have been used directly, we prefered to perform an independent validation using the Tellurium system for modeling (Choi et al, 2018), which also allows flexibility in the choice of initial conditions and parameters. Steady-state

values obtained from the formulas obtained with Maxima were fed into a Tellurium model that can be described by the following expressions:

*model rnadeg_deadenylation_dependent*

$$\$T \rightarrow PAl; \ kS * T$$

$$PAl \rightarrow PAm; \ kA * PAl$$

$$PAm \rightarrow PAs; \ kA * PAm$$

$$PAs \rightarrow \ ; \ kD * PAs$$

and

*model rnadeg_deadenylation_independent*

$$\$T \rightarrow PAl; \ kS * T$$

$$PAl \rightarrow PAm; \ kA * PAl$$

$$PAl \rightarrow \ ; \ kD * PAl$$

$$PAm \rightarrow PAs; \ kA * PAm$$

$$PAm \rightarrow \ ; \ kD * PAm$$

$$PAs \rightarrow \ ; \ kD * PAs$$

Variations in $k_D$ and $k_A$ values were used to follow the change in the simulated species amounts of RNA over time, in the absence of new RNA generation (models missing the generation of PAl). The sum of the three species and its decrease over time were analyzed similar to how real experimental data are processed to obtain estimates of half-life. The PAl, PAm, and PAs were assigned arbitrary values of 70, 40, and 10 nucleotides, to be able to calculate average simulated poly(A) tail length over time.

## Data availability

The accession numbers for the sequencing data reported in this paper are GSE160642 for total and purified RNA and GSE247954 for SLAM-seq (GEO, Gene Expression Omnibus, NCBI). The MS proteomics data that support the findings of this study have been deposited in the ProteomeXchange repository with the dataset identifier PXD028008. Nanopore sequencing results were deposited in GEO, with accession number GSE211782.

The source data of this paper are collected in the following database record: biostudies:S-SCDT-10_1038-S44318-024-00250-x.

## Peer review information

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

## Acknowledgements

We thank our colleagues of the Genetics of Macromolecular Interactions laboratory, in particular Micheline Fromont-Racine and Alain Jacquier for fruitful discussions and support and Lucia Oreus for assistance in media and reagents preparation. We are grateful to Guilhem Janbon and our colleagues of the RNA Biology of Fungal Pathogens laboratory for support and discussions. We thank Mariette Matondo and Julia Chamot-Rooke for providing access to the Orbitrap Velos mass spectrometer at the Proteomics facility of the Institut Pasteur. We acknowledge the assistance of Laurence Ma (Biomics Platform, C2RT, Institut Pasteur, Paris) for DNA sequencing. We are grateful to G Bushkin, J Coller, D Challal and D Libri for sharing reagents and strains. French Agence Nationale de la Recherche grant ANR-18-CE11-0003 (CS). French Ministère de l'Enseignement supérieur, de la Recherche et de l'Innovation PhD grant (LA). Fondation ARC pour la Recherche sur le Cancer (LA). Swiss National Science Foundation Advanced Grant TMAG-3_209354 (KW).

## Author contributions

**Léna Audebert**: Conceptualization; Data curation; Formal analysis; Funding acquisition; Validation; Investigation; Visualization; Methodology; Writing—original draft; Writing—review and editing. **Frank Feuerbach**: Conceptualization; Resources; Supervision; Validation; Investigation; Visualization; Methodology; Writing—original draft. **Mostafa Zedan**: Resources; Methodology; Writing—review and editing. **Alexandra P Schürch**: Resources; Methodology; Writing—review and editing. **Laurence Decourty**: Resources; Data curation; Validation; Investigation; Methodology; Writing—review and editing. **Abdelkader Namane**: Data curation; Formal analysis; Validation; Investigation; Visualization; Methodology; Writing—review and editing. **Emmanuelle Permal**: Formal analysis; Writing—review and editing. **Karsten Weis**: Conceptualization; Supervision; Funding acquisition; Investigation; Writing—review and editing. **Gwenaël Badis**: Data curation; Formal analysis; Validation; Investigation; Visualization; Methodology; Writing—review and editing. **Cosmin Saveanu**: Conceptualization; Resources; Data curation; Formal analysis; Supervision; Funding acquisition; Validation; Investigation; Visualization; Methodology; Writing—original draft; Project administration; Writing—review and editing.

Source data underlying figure panels in this paper may have individual authorship assigned. Where available, figure panel/source data authorship is listed in the following database record: biostudies:S-SCDT-10_1038-S44318-024-00250-x.

## Disclosure and competing interests statement

The authors declare no competing interests.

# Expanded View Figures

**Figure EV1. Enrichment of specific factors in purified complexes containing Upf1.**

Volcano plots represent the enrichment of proteins, relative to the protein abundance in a total extract as measured by label-free mass spectrometry. Results are presented for purifications using TAP tagged versions of Lsm1 (**A**), Pat1 (**B**), Lsm7 (**C**), and Dhh1 (**D**). Ribosomal proteins are highlighted in blue. (**E**) Examples of enrichment values across the various purifications for Lsm2, a component of Lsm1-7 and Lsm2-8 complexes, and Upf1. Error bars represent standard deviation. *N* is 3 for Dhh1, Lsm1, Lsm7, and Pab1 purifications and 4 for Pat1.

▶

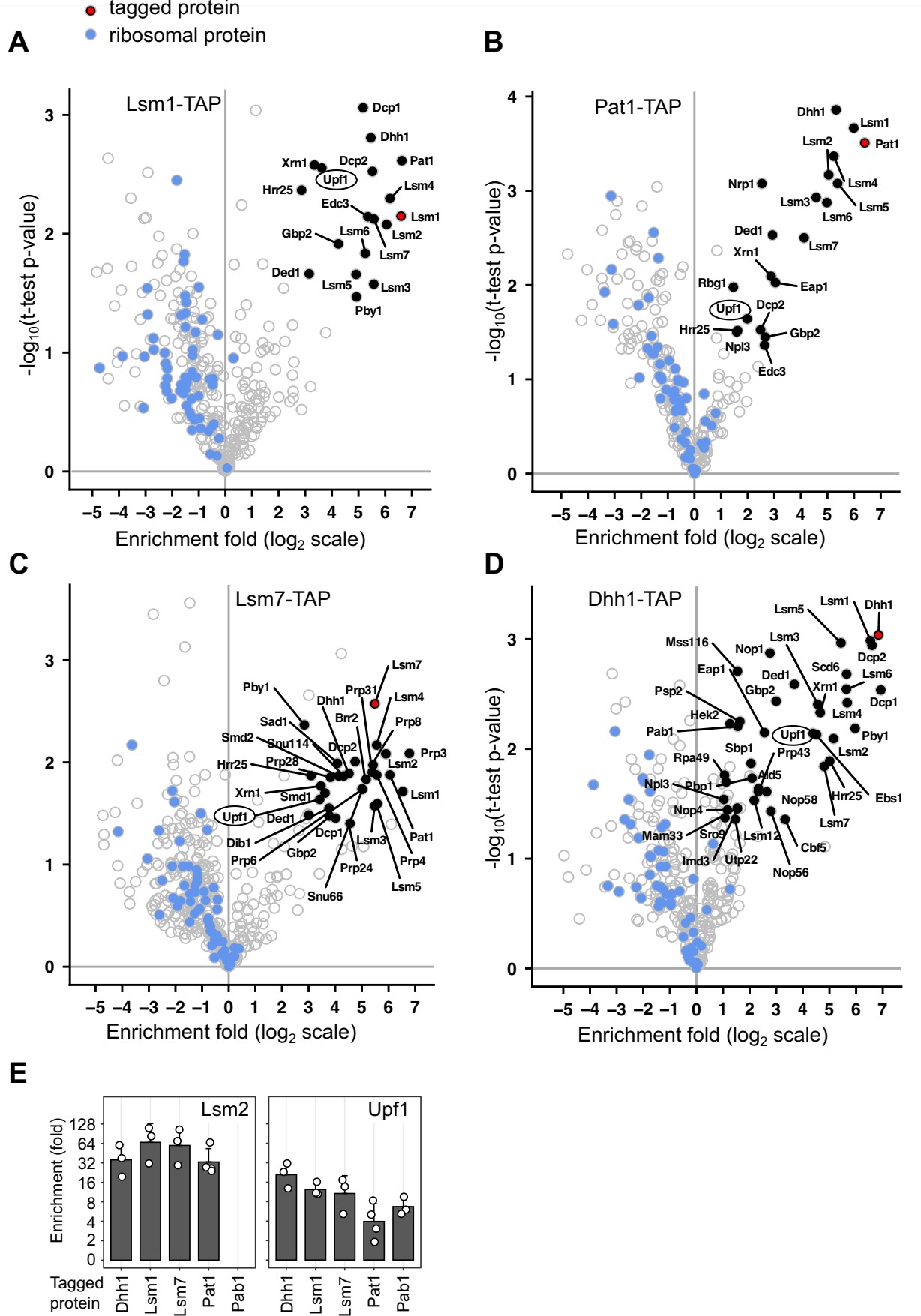

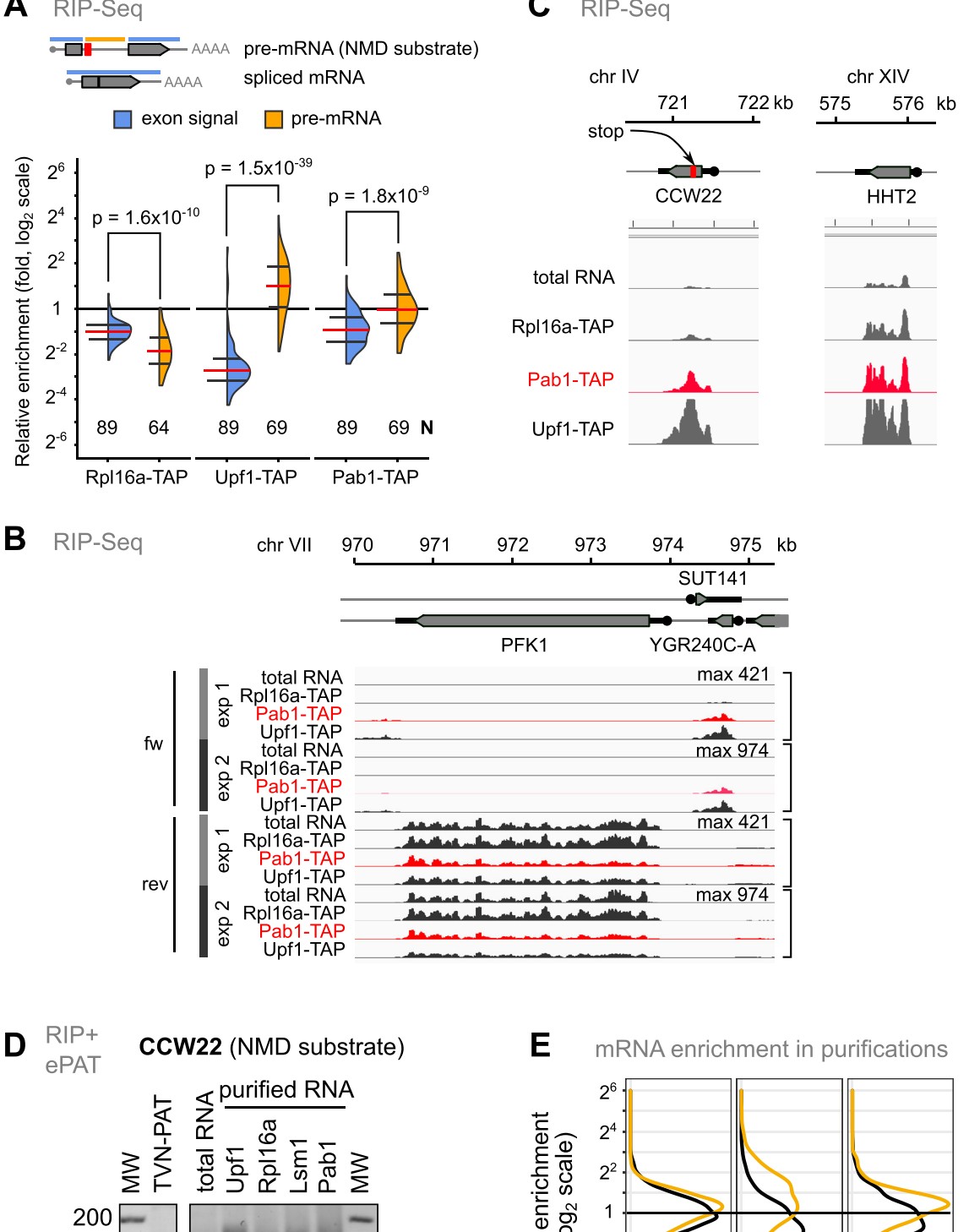

◀ **Figure EV2. Unstable mRNAs are associated with Pab1.**

(A) Relative enrichment of pre-mRNA for ribosomal protein genes (intron signal, orange) in comparison with the spliced mRNA for the same category (exon signal, blue). The red horizontal line indicates the median of the enrichment values, with the first and third quartiles indicated. The p-values correspond to a Welch two sample t-test, *N* is the number of values in each category. (B) Example of signal intensity for RNAs associated with Pab1-TAP, Rpl16a-TAP, or Upf1-TAP for a region of the yeast genome that corresponds to SUT141, an unstable RNA, in comparison with the PFK1 stable mRNA (transcribed in proximity but from the opposite strand). The vertical scale is the same for the samples of the same replicated experiment. Pab1 associated RNA signal is depicted in red. The image was obtained with the Integrated Genome Viewer, IGV. (C) Similar to (B) for the regions of transcripts CCW22, and NMD substrate (left), and HHT2 (right). (D) Estimation of poly(A) tail distribution for CCW22 and HHT2 in the RNA fractions enriched with Upf1, Rpl16a, Lsm1, and Pab1. The TVN-PAT lane corresponds to a control experiment with an anchored 3′ primer. For CCW22, two 3′ ends were detected (noted A12, and A12′). (E) The distribution of the enrichment values for two extreme categories of mRNA, with long (orange) or short average poly(A) tails (black) was depicted for the purified samples in association with Rpl16a-TAP, Upf1-TAP, and Pab1-TAP. A Wilcoxon ranks sum test with continuity correction comparing the distribution of relative enrichment values for long poly(A) tails mRNA versus short poly(A) tails, led to the following p values (for a minimum 25% increase): 0.9997 for Rpl16a, less than $2 \times 10^{-16}$ for Upf1 and $1.9 \times 10^{-8}$ for Pab1.

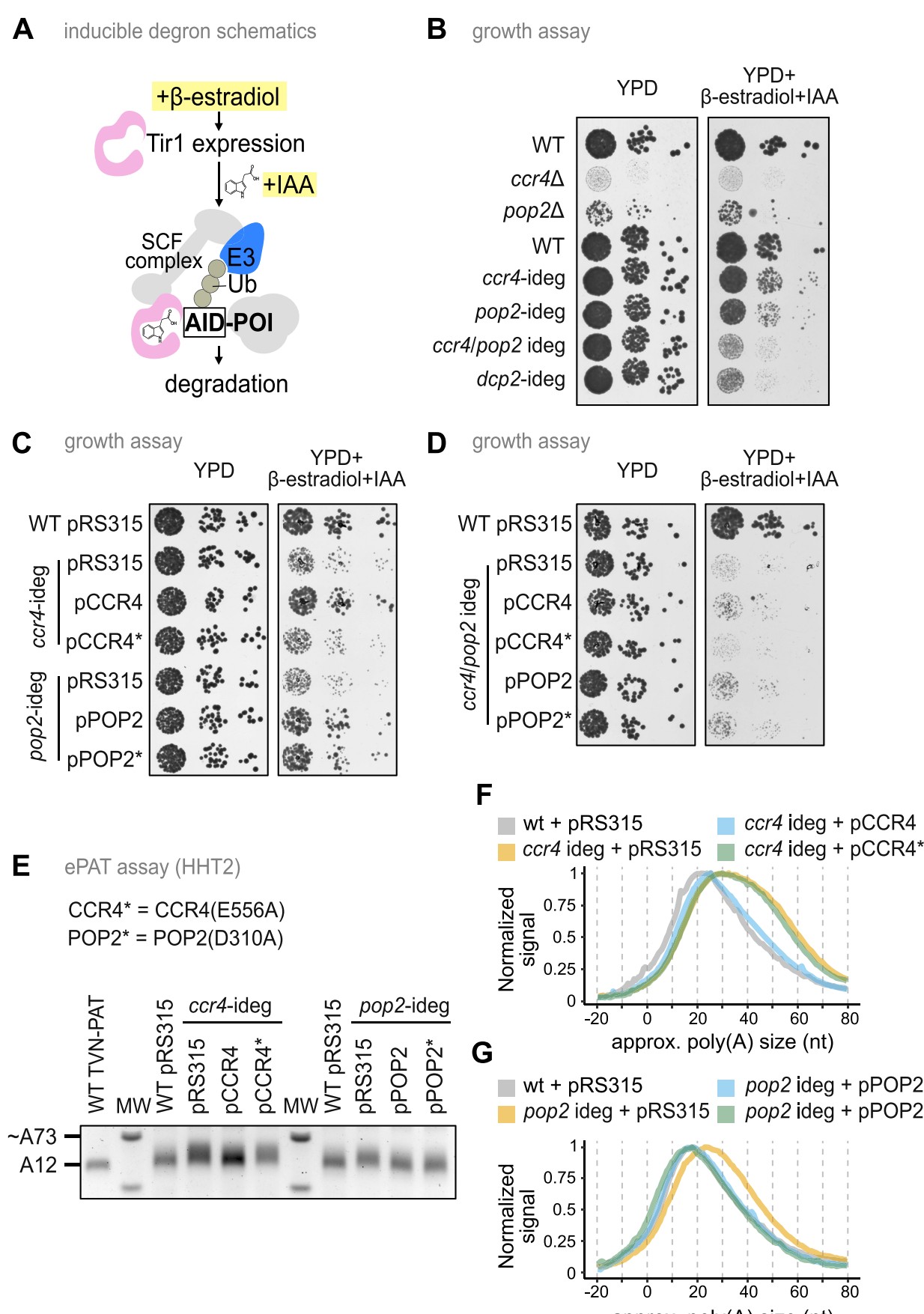

**A** inducible degron schematics

**B** growth assay

**C** growth assay

**D** growth assay

**E** ePAT assay (HHT2)

CCR4* = CCR4(E556A)
POP2* = POP2(D310A)

**F**

**G**

**Figure EV3. An inducible degron system for deadenylases.**

(A) Schematics of the developed inducible degron system in which *O. sativa* Tir1 expression is under the control of an estrogen sensitive promoter. In the presence of the plant auxin hormone indole-3-acetic acid (IAA), Tir1 targets a E3 ubiquitin ligase to the AID domain fused to the protein of interest (POI) for its rapid proteasome degradation. (B) Serial dilution plate growth assay for deletion and inducible degron (ideg) strains affecting deadenylases of the CCR4/NOT complex and the decapping enzyme Dcp2. Growth on rich medium (YPD, left) was compared with growth on rich medium containing β-estradiol and IAA (right). (C) Similar to (B) for the strains depleted for Ccr4 or Pop2 that were transformed with a control plasmid (pRS315), pRS315-CCR4 or pRS315-POP2. The asterisks indicates mutated versions of Ccr4, E556A, and Pop2, D310A. (D) Similar with (C), but for the concomitant depletion of Ccr4 and Pop2. (E) Changes in the poly(A) tail size of HHT2 measured using ePAT in the strains presented in panel (C). The TVN-PAT lane indicates the relative position of the A12 RNA. (F) Poly(A) size profile for HHT2 under Ccr4 depletion conditions was extracted from panel (E) and normalized to the signal in each lane. (G) Similar to (F), but for the depletion of Pop2.

**A** Northern blot

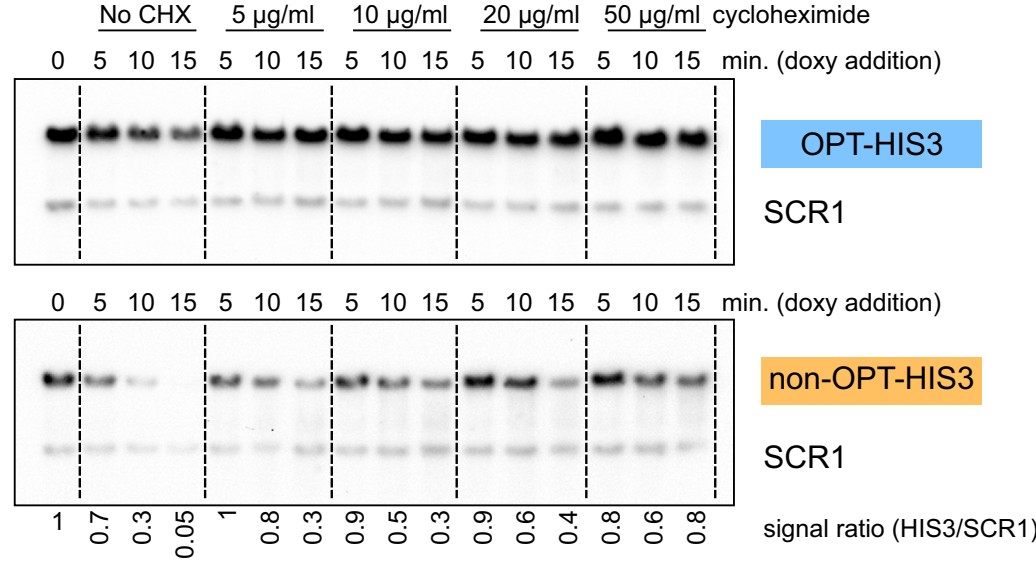

**B** wt

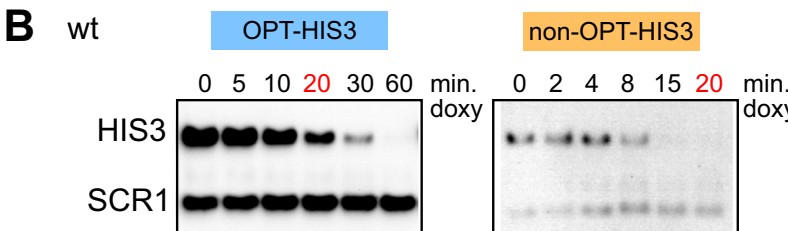

**C** *upf1*-ideg

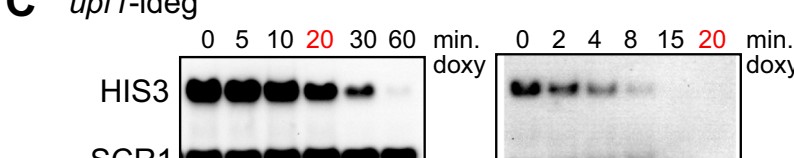

**D** *ccr4*-ideg

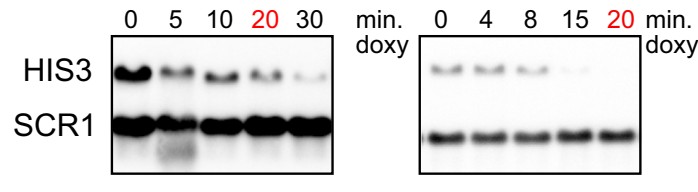

half-life 12.6 +/- 3.3 min          half-life 8.8 +/- 2.6 min

**E** *pop2*-ideg

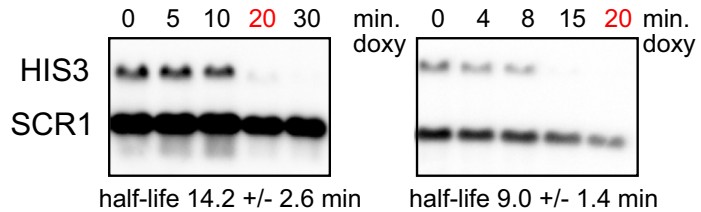

half-life 14.2 +/- 2.6 min          half-life 9.0 +/- 1.4 min

◄ **Figure EV4.** **Degradation of OPT-HIS3 and non-OPT-HIS3 reporter mRNAs depends on translation and decapping.**

(A) Dose-dependent effect of the translation inhibitor cycloheximide (from 0 to 50 μg/ml) was tested by Northern blotting for optimal (upper panel) and non-optimal (lower panel) HIS3 reporters. SCR1 was used as a loading control. Degradation rates for optimized HIS3 (left panel) and non-optimal HIS3 (right panel) were visualized by Northern blot (HIS3 reporter signal) in comparison with an SCR1 control. (B) Similar to (A), in a wild type strain. (C) Similar to (A) after depletion of Upf1. (D) Similar to (A) after depletion of Ccr4. (E) Similar to (A) after depletion of Pop2. For each situation, the estimated half-life and the 95% confidence interval obtained from RT-qPCR and the results are indicated.

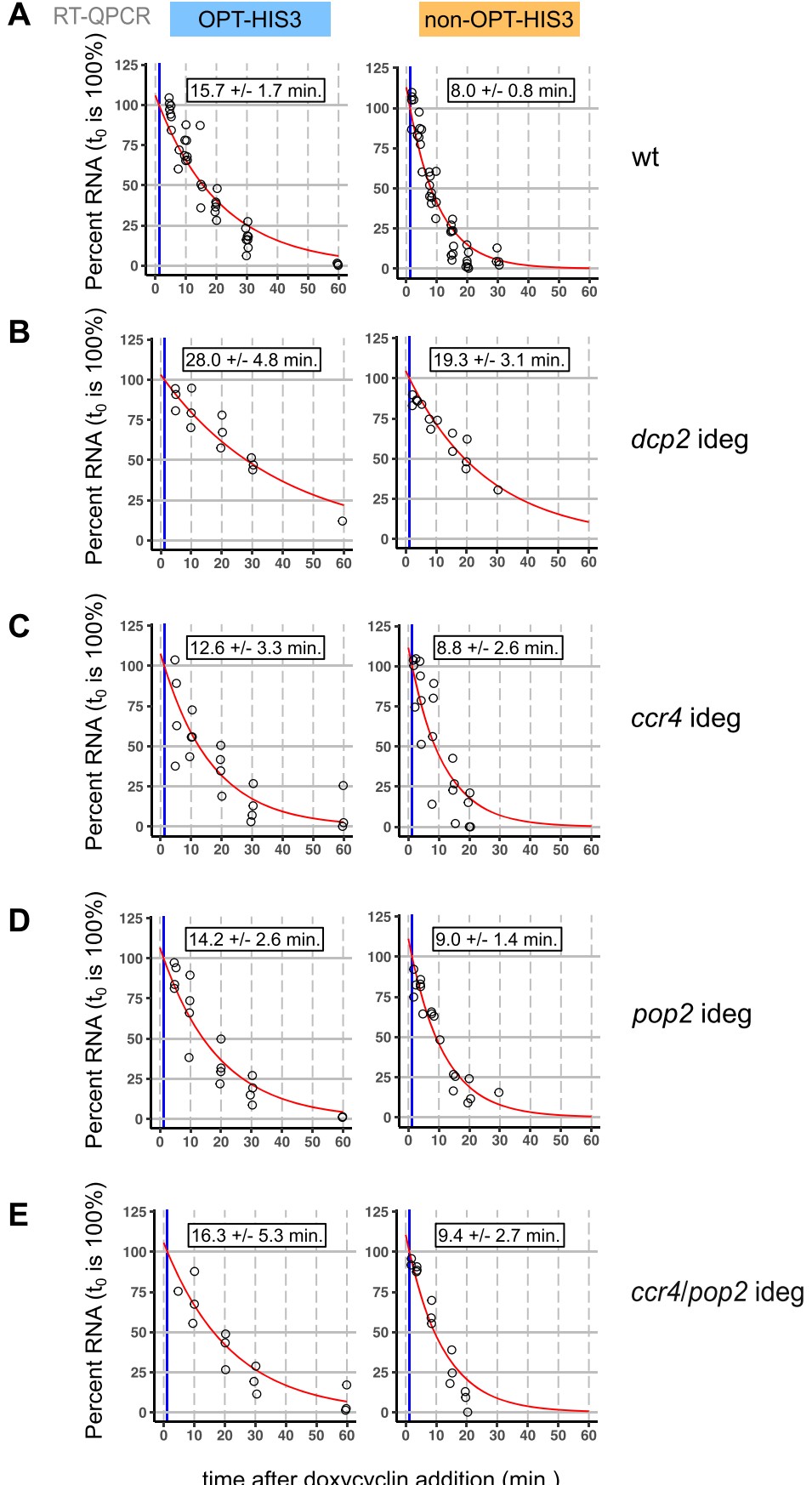

time after doxycyclin addition (min.)

◄ **Figure EV5.  Half-life of reporter RNAs as measured by RT-qPCR.**

(**A**) The half-life of HIS3 reporter, codon-optimized (left panel) or non-optimized (right panel), was measured by RT-qPCR following transcription block with doxycyclin. For each decay experiment, the values were normalized ot time 0, considered to represent 100%. Indicated estimates correspond to half-life in minutes with a 95% confidence interval. All the experiments were performed independently at least three times. (**B**) Similar to (**A**), but after depletion of Dcp2. (**C**) Similar to (**A**), but after depletion of Ccr4. (**D**) Similar to (**A**), but after depletion of Pop2. (**E**) Similar to (**A**) but after concomitant depletion of Ccr4 and Pop2.

