## [Peer Review File · The EMBO Journal]

RNA degradation triggered by decapping is largely independent of initial deadenylation

Léna Audebert, Frank Feuerbach, Mostafa Zedan, Alexandra Schürch, Laurence Decourty, Abdelkader Namane, Emmanuelle Permal, Karsten Weis, Gwenael Badis, and Cosmin Saveanu

Corresponding author(s): Cosmin Saveanu (cosmin.saveanu@pasteur.fr)

Review Timeline:

Submission Date:	11th Jan 24
Editorial Decision:	30th Mar 24
Revision Received:	19th Jul 24
Editorial Decision:	12th Aug 24
Revision Received:	29th Aug 24
Accepted:	11th Sep 24

Editor: Cornelius Schneider

Transaction Report:

Dear Dr. Saveanu,

Thank you again for submitting your manuscript to the EMBO Journal and for sharing your preliminary point-by-point reply. We have now read your response letter and we find the additional experiments and arguments helpful and productive. We would therefore like to invite you to submit a revised version of the manuscript, addressing the comments of all three reviewers.

I should also add that it is The EMBO Journal policy to allow only a single major round of revision and that it is therefore important to resolve the main concerns at this stage.

We generally allow three months as standard revision time, which can be extended to 6 months in case of major revisions, such as the experiments required here. As a matter of policy, competing manuscripts published during this period will not negatively impact on our assessment of the conceptual advance presented by your study. However, we request that you contact the editor as soon as possible upon publication of any related work, to discuss how to proceed. Should you foresee a problem in meeting the deadline, please let us know in advance and we may be able to grant an extension.

Thank you for the opportunity to consider your work for publication. I look forward to your revision.

Yours sincerely,

Cornelius Schneider

Cornelius Schneider, PhD
Editor
The EMBO Journal
c.schneider@embojournal.org

Please remember: Digital image enhancement is acceptable practice, as long as it accurately represents the original data and

conforms to community standards. If a figure has been subjected to significant electronic manipulation, this must be noted in the figure legend or in the 'Materials and Methods' section. The editors reserve the right to request original versions of figures and the original images that were used to assemble the figure.

We realize that it is difficult to revise to a specific deadline. In the interest of protecting the conceptual advance provided by the work, we recommend a revision within 3 months (28th Jun 2024). Please discuss the revision progress ahead of this time with the editor if you require more time to complete the revisions. Use the link below to submit your revision:

Referee #1:

In this manuscript, Audebert and colleagues set out to critically examine the long-held assumption that mRNA deadenylation is a general prerequisite for mRNA decapping and decay. The authors start their manuscript by providing a comprehensive introduction describing the data supporting and opposing this assumption. Next they analyze own and published data showing that unstable mRNAs have indeed long poly(A) tails. The authors model the classical deadenylation-dependent and a new proposed deadenylation-independent model and show that to explain the genomic data there is need for decapping to occur even in mRNAs with long poly(A) tails. To test this model experimentally, the authors use an Auxin inducible degron of the main deadenylases followed nanopore sequencing to measure poly(A) tails and SLAM-Seq to measure mRNA stability. Finally, they use a reporter system to show that changes in the poly(A) length of the reporter does not affect the mRNA stability of the reporter.

Overall, I think this is a very solid manuscript with a very balanced discussion. It brings new data to discuss a classical assumption of RNA biology that will be important for the whole community. I do not have any major concern that would prevent its publication.

Referee #2:

The consensus model of general mRNA turnover in eukaryotes, largely based on studies of the turnover of reporter mRNAs following transcription inhibition in both yeast and mammalian cells, is that mRNA turnover predominantly involves an initial slow deadenylation phase followed by decapping and subsequent 5' degradation. The same approaches have shown that some NMD transcripts are rapidly degraded in a largely deadenylation-independent manner. More recently, broader studies have demonstrated a strong correlation between poly(A) tail length and mRNA instability, with transcripts of stable house keeping genes having predominantly short poly(A) tails at steady state. In this manuscript, the authors propose that a significant fraction of mRNA degradation occurs in yeast in a "deadenylation-independent" manner. This model contravenes the consensus deadenylation-dependent model and so is controversial, potentially highly significant and of interest to a broad readership. Upf1 and Pab1 were found to be co-enriched with NMD substrates that have long poly(A) tails at steady state. Consideration of published datasets led to the conclusions that Pab1 is associated with mRNAs with longer poly(A) tails, whether they are NMD substrates or not, that poly(A) tail length is inversely correlated with stability and that slowing deadenylation rates did not cause changes in poly(A) tail length that are consistent with alterations in stability. The authors then modelled mRNA turnover at different rates of deadenylation and decapping and concluded that changes in the decapping rate, rather than the rate of deadenylation, might explain the correlation between poly(A) tail length and instability. Depletion of both deadenylases caused an increase in poly(A) tail length while depletion of Dcp2 caused a decrease in poly(A) tail length. Notably, poly(A) tail shortening upon Dcp2 depletion was coupled to mRNA accumulation, while this was not seen upon depletion of the deadenylases. Similar findings were reported using transcription-repressible reporter constructs. SLAM-seq experiments in cells depleted for Dcp2 or the Ccr4 and Pop2 deadenylases revealed distinct alterations in mRNA stabilities, with different transcripts being differentially sensitive to perturbations in decapping or deadenylation. The authors point out that these observations are not consistent with a linear deadenylation and decapping pathway. Analysis of reporter mRNA poly(A) tail lengths upon depletion of either Ccr4 or Pop2 revealed increases in the longest poly(A) tails and depletion of the short poly(A) tails. Based on these findings, the authors propose a predominantly deadenylation-independent degradation pathway for unstable mRNAs.

The results broadly support the conclusion that the current consensus "deadenylation then decapping" model of bulk mRNA turnover is overly simplistic, particularly for mRNAs with relatively short half-lives. The experimentation is technically of a high quality and statistically robust. As commented by the authors, the data can be interpreted either in a deadenylation-dependent manner involving a rate-limiting decapping step or by a model where the prerequisite for deadenylation to an oligoadenylate tail before rapid degradation no longer holds. The phrase "deadenylation-independent degradation" (used in the text and implied in the title) is used to describe scenarios where rapid degradation is triggered before the poly(A) tail is shortened to an oligoadenylate length. I find this misleading, as much of the data supports a role for the deadenylases in shortening poly(A) tails

during mRNA degradation. I think it would be more appropriate to describe this in terms of an uncoupling of the decapping and deadenylation events, as stated in the abstract.

Specific comments

Pab1 is shown to be enriched with NMD substrates, while mRNAs with long poly(A) tails are reported to be enriched in the Pab1-bound fraction (EV2C). I am not convinced from the data shown that the latter is a fair conclusion, as the traces for longer poly(A) mRNAs (in yellow) look comparable in the Rpl16a and Pab1 enriched fractions. "Other" RNAs are apparently not enriched in the Pab1-bound fraction (1C). Does this mean that the highly expressed house-keeping genes that typically have short poly(A) tails, such as ribosomal protein encoding transcripts, are not enriched in the Pab1-bound fraction? If this is the case, could it be that Pab1 enrichment is dependent upon multiple copies of Pab1 bound to longer poly(A) tails? Or that enrichment is less sensitive to highly expressed transcripts?

The authors state that the most stable and least stable mRNAs showed opposite changes in poly(A) tail length in deadenylase mutants (p4, Fig S1C) but, from the data, it appears that both classes of transcript show a positive (increased) change in poly(A) tail length compared to the wild-type strain (S1C). Since changes in poly(A) tail length upon depletion of the deadenylases does not correlate with an increase in stability, the authors question "the impact of deadenylation on RNA degradation in yeast". The data supports the conclusion that deadenylation occurs but is not rate-limiting - there appears to be an impact when depletion of the enzymes affects poly(A) tail length and the poly(A) tails shorten over time.

The computational modelling data lead the authors to conclude that decapping rates may underlie the observed correlation between poly(A) tail length and mRNA instability, as a slow decapping rate is required to observe oligoadenylated mRNAs. I assume a deadenylation rate of 0.25/min relates to conversion of PAI to PAm or PAm to PA0. Is the "high" decapping rate of 0.5 min⁻¹ based on experimental data? This seems similar to that measured for MFA2 reporters in Roy Parker's lab but is considerably lower than that observed in *in vitro* decapping assays (e.g. Wurm et al., 2017). Would modelling using higher decapping rates have a substantial effect?

Why does the *ccr4Δ* mutant have such a poor growth phenotype on YPD medium, compared to the wild-type strain (Fig EV3)? The slow growth of strains depleted for deadenylases in the auxin degron strains might suggest an important role for deadenylation in cell growth. These seems to be a stronger effect than observed for *upf1Δ* mutants that block NMD.

An increase in poly(A) tail length upon depletion of the Ccr4 and Caf1 deadenylases (3B, EV5) suggests that the mRNAs are subjected, at least to a degree, to initial deadenylation. If a deadenylation pathway is in a simple competition with a decapping-mediated pathway, then depletion of the deadenylases would stimulate decapping and the poly(A) tail lengthening effect would be dampened, particularly for unstable mRNAs with a high decapping rate. This data may therefore support a deadenylation-dependent initial phase within the pathway.

The data in 3C and 3D suggests that depletion of Dcp2, but not depletion of the deadenylases, causes a block in turnover (supported by data in Fig 5). If decapping rather than deadenylation were rate-limiting for an average mRNA, then I think this is what you would expect even if deadenylation preceded decapping.

It would perhaps be insightful to deplete Dcp2 and the decapping enzymes together. This would address whether other pathways are involved to a significant degree.

It's not obvious why data is shown for the OPT-HIS3 and non-OPT-HIS3 reporters when the analysis of an unstable mRNA with a long poly(A) tail and a stable, house keeping transcript would have been more in keeping with other experiments shown.

it would be useful to show the wild-type poly(A) profiles for comparison in Fig EVC,D and stack the panels, as in Fig. 5G.

I find the data in Fig EV5 quite informative as this data allows a direct comparison of poly(A) tail lengths during mRNA turnover. I would propose to include this in Fig 5 (Fig 5 panels A, B and F could be put in the EV figure).

Referee #3:

This manuscript explores the role of deadenylation in stimulating decapping. Based on decades of studies, one of the paradigms in the mRNA decay field has been that deadenylation precedes (and is required for) decapping. Here, the authors put forward a different model, where decapping can occur independently of deadenylation. This model is based on analysis of mRNA half-lives, poly(A)-tail lengths, and changes in both upon depletion of components of the deadenylase or decapping machineries. Recent studies have demonstrated that there is substantially more subtlety in these relationships than the field would have initially been predicted in the 1990s, but the conclusions of this manuscript are still surprising - where decapping occurs independently of deadenylation and is the major determinant of mRNA stability. Clearly, the role of deadenylation and decapping are still hotly debated - and from this perspective, this paper will be of interest - however, as it stands, the authors make many unstated assumptions that may (likely) be the reason their conclusions differ so much from the established framework of mRNA decay. Thus, without substantially more experimentation and analysis, the interpretations of results and the conclusions are

over-stated without proper discussion of the published literature, and I cannot recommend publication in its current form.

Major comments:

1. To conclude that "Pab1 preferentially binds long poly(A)-tail RNAs" needs more evidence. For instance, the authors need to show that RNAs bound by Pab1 have longer poly(A) tails than those originating from the same gene and not bound by Pab1. Given that this analysis and other hand-picked examples of gene classes are the bulwark of their conclusion that "the association of an mRNA with Pab1 can be considered a marker of their instability," substantially more experiments need to be performed to exclude other alternative explanations.
2. The authors mention Eisen, et al 2020, but do not discuss the fundamental differences in their models from the published one: here, the authors assume that decapping rate is constant with different poly(A)-tail lengths (for transcripts from the same gene), which is opposite to the conclusions of Eisen et al. (which found that the two rates, in mammalian cells, are often strongly correlated and decapping rates are not constant). This assumption appears to be the basis of the difference in conclusions between the two papers. Indeed, the previously published paper explained the lack of correlation between tail lengths and stability to be partly due to this fundamental link between deadenylation and decapping rates. How do the authors' models and conclusions change if they do not assume the same decapping rates for different tail lengths? How can the authors exclude this alternative model?
3. The authors deplete various deadenylase components that could also impact the deadenylation-decapping coupling implied by Eisen et al, papers from the Izaurralde group, etc. What happens when catalytic mutants are added back? Can the authors distinguish between the catalytic aspect of deadenylation vs a structural requirement for the deadenylase complexes?

Audebert et al., answers and comments to the major points raised by reviewers

We thank reviewer 1 for the words of appreciation about the results presented in the manuscript.

Answers to questions from reviewer 2

***Question 1:** The results broadly support the conclusion that the current consensus "deadenylation then decapping" model of bulk mRNA turnover is overly simplistic, particularly for mRNAs with relatively short half-lives. The experimentation is technically of a high quality and statistically robust. As commented by the authors, the data can be interpreted either in a deadenylation-dependent manner involving a rate-limiting decapping step or by a model where the prerequisite for deadenylation to an oligoadenylate tail before rapid degradation no longer holds. The phrase "deadenylation-independent degradation" (used in the text and implied in the title) is used to describe scenarios where rapid degradation is triggered before the poly(A) tail is shortened to an oligoadenylate length. I find this misleading, as much of the data supports a role for the deadenylases in shortening poly(A) tails during mRNA degradation. I think it would be more appropriate to describe this in terms of an uncoupling of the decapping and deadenylation events, as stated in the abstract.*

Answer 1: We agree that the "deadenylation-independent degradation" terminology is not fully consistent with the fact that deadenylation is part of the RNA degradation process and it is indeed the lack of coupling and the possibility of both deadenylation and decapping occurring in parallel that is what we investigated. We thus removed from the text any mention of "deadenylation-independent degradation". For example, the original phrasing:

"Slowing down deadenylation is expected to lead first to the accumulation of unstable RNAs in the classical model and to have no impact in deadenylation-independent RNA degradation..."

now reads:

"Slowing down deadenylation is expected to lead first to the accumulation of unstable RNAs in the classical model and to have no impact if initiation of RNA degradation is not dependent on prior deadenylation..."

and

"Thus, perturbation of deadenylation or decapping led to results that are compatible with a deadenylation-independent degradation of unstable RNAs in yeast."

now reads:

"Thus, perturbation of deadenylation or decapping led to results that are compatible with an uncoupling between these processes during the degradation of unstable RNAs in yeast."

We agree that a different terminology could be used for the two models for the kinetics of RNA degradation, one in which initial deadenylation is a requirement for decapping, and one in which it is not. For example, naming them "serial" (deadenylation, then decapping) and "parallel" (deadenylation and decapping occurring uncoupled) could have helped, but would make the link to the known pathways and previous publications ("deadenylation-dependent" and "deadenylation-independent") less obvious. We reformatted the text in the revised version to try to keep the terminology consistent.

Question 2: *Pab1 is shown to be enriched with NMD substrates, while mRNAs with long poly(A) tails are reported to be enriched in the Pab1-bound fraction (EV2C). I am not convinced from the data shown that the latter is a fair conclusion, as the traces for longer poly(A) mRNAs (in yellow) look comparable in the Rpl16a and Pab1 enriched fractions. "Other" RNAs are apparently not enriched in the Pab1-bound fraction (1C). Does this mean that the highly expressed house-keeping genes that typically have short poly(A) tails, such as ribosomal protein encoding transcripts, are not enriched in the Pab1-bound fraction? If this is the case, could it be that Pab1 enrichment is dependent upon multiple copies of Pab1 bound to longer poly(A) tails? Or that enrichment is less sensitive to highly expressed transcripts?*

Answer 2: To answer the question of whether Pab1 specifically binds to long poly(A) tail transcripts and if these are truly enriched in the purified fraction together with Pab1 we performed a control experiment, now shown in **Fig. EV2C, D** and additional statistical tests, mentioned in the legend to **Fig. EV2E** (former EV2C mentioned by the reviewer). We used a non-parametric test to estimate to what extent there was a difference between the distribution of relative enrichment values for the two categories of transcripts (short and long average poly(A) tail. The Wilcoxon rank sum test using the null hypothesis that the two distributions have the same location led to a p-value of 6.5×10^{-5} for the Rpl16a situation and values lower than 2×10^{-16} for the Upf1 and Pab1 conditions. We thus used the Wilcoxon test with a different and more stringent null hypothesis, that the shift in the location of the two distributions is larger than 25%. The new legend to **Fig. EV2E**, including the obtained results reads:

"(E) The distribution of the enrichment values for two extreme categories of mRNA, with long (orange) or short average poly(A) tails (black) was depicted for the purified samples in association with Rpl16a-TAP, Upf1-TAP and Pab1-TAP. A Wilcoxon ranks sum test with continuity correction comparing the distribution of relative enrichment values for long poly(A) tails mRNA versus short poly(A) tails, led to the following p values (for a minimum 25% increase): 0.9997 for Rpl16a, less than 2×10^{-16} for Upf1 and 1.9×10^{-8} for Pab1."

In addition to these additional statistical considerations, we wanted to show, by an independent method, that specific transcripts enriched with Pab1 had longer poly(A) tails. Consistent with the reviewer suggestion, we would expect that the presence of several Pab1 molecules on a longer poly(A) tail would strengthen the interaction and thus the ability of these transcripts to be purified with Pab1. We used ePAT, a method that allows an estimation of poly(A) tail length, to show, first, that an NMD substrate has relatively long poly(A) tails both in a total extract and in association with Pab1 (new panels EV2C and D). Next, we looked at the abundant transcript for the histone Hht2 which has, on average, short poly(A) tails. The fraction of this transcript associated with Pab1 showed an extended poly(A) tail, not visible in the total RNA extract. Thus, the purification of RNAs associated with Pab1 is particularly good at enriching long poly(A) tail transcripts and allows the detection of populations of RNA that are not visible in a total extract. For convenience, panel D and the corresponding legend are included here (**Fig. Reviewer 2/Answer 2**).

Question 3: *The authors state that the most stable and least stable mRNAs showed **opposite** changes in poly(A) tail length in deadenylase mutants (p4, Fig S1C) but, from the data, it appears that both classes of transcript show a positive (increased) change in poly(A) tail length compared to the wild-type strain (S1C). Since changes in poly(A) tail length upon depletion of the deadenylases does not correlate with an increase in stability, the authors question "the impact of deadenylation on RNA degradation in yeast". The data supports the conclusion that deadenylation occurs but is not rate-limiting - there appears to be an impact when depletion of the enzymes affects poly(A) tail length and the poly(A) tails shorten over time.*

Answer 3: We agree with the reviewer that the term "opposite" was probably misused here and amended the text accordingly.

Initial version:

"Most stable 25% RNAs and most unstable 25% RNAs showed opposite relative changes in poly(A) tail length (Fig. S1C)."

Revised version:

"Most stable 25% RNAs and most unstable 25% RNAs showed different relative changes in poly(A) tail length, with a higher relative increase in the poly(A) tail length for relatively stable transcripts in one case, deletion of PAN2, and with a lower effect on this class of transcripts in the absence of CCR4 (Appendix Fig. S1C)." - page 4

D RIP+
ePAT **CCW22 (NMD substrate)**

Figure Reviewer 2/Answer 2

Fig. EV2(D) Estimation of poly(A) tail distribution for CCW22 and HHT2 in the RNA fractions enriched with Upf1, Rpl16a, Lsm1 and Pab1. The TVN-PAT lane corresponds to a control experiment with an anchored 3' primer. For CCW22, two 3' ends were detected (noted A12, and A12').

Question 5: The computational modelling data lead the authors to conclude that decapping rates may underlie the observed correlation between poly(A) tail length and mRNA instability, as a slow decapping rate is required to observe oligoadenylated mRNAs. I assume a deadenylation rate of 0.25/min relates to conversion of PAI to PAm or PAm to PA0. Is the "high" decapping rate of 0.5 min⁻¹ based on experimental data? This seems similar to that measured for MFA2 reporters in Roy Parker's lab but is considerably lower than that observed in in vitro decapping assays (e.g. Wurm et al., 2017). Would modelling using higher decapping rates have a substantial effect?

Answer 5: We used as test values the previously evaluated decapping rates for oligoadenylated RNA, in the model of Cao and Parker, *RNA* 2001, PMID 11565744), which were approximately 0.46 min⁻¹ for the rapidly degraded MFA2 transcript and 0.03 min⁻¹ for the stable PGK1 transcript. These were computationally derived from experimental data. Deadenylation rates, probably more suitable called pseudo-rates, were estimated in the Cao and Parker lab to vary between 1.3 min⁻¹ for MFA2 and 0.024 min⁻¹ for PGK1 for steps corresponding to the loss of 10 A. They would thus correspond to something between 0.002 and 0.1 A/min. The values we used for the examples presented in Fig. 2E correspond to a range of 0.0003 to 0.08 A/min., which are in a comparable range.

The much higher values measured for Dcp2 complexes for decapping in vitro (Wurm et al. 2017), ranging in the 10-30min⁻¹ can be included in the model. For the one in which deadenylation preceded decapping, having much higher values for the decapping rate leads to a marked decrease in the steady-state levels of the shorter poly(A) forms. However, these forms are already minor at the highest decapping rate that we initially used for modeling and the overall stability of RNA would be only

slightly decreased (deadenylation being limiting). For the parallel degradation model, the situation is different, as in this case, having much faster decapping would lead to a shorter half-life, characterized also by mostly long poly(A) tails (no time for deadenylation).

Since these informations can be useful for the manuscript we added the following in the revised version:

"The values of the rate constants used in these simulations were in the range of those estimated previously from experimental and kinetic modeling results (Cao & Parker, 2001), even if we are aware that measured *in vitro* decapping rates can be 10 to 50 times higher (Wurm et al, 2017)." - page 5

Question 6: Why does the *ccr4*Δ mutant have such a poor growth phenotype on YPD medium, compared to the wild-type strain (Fig EV3)? The slow growth of strains depleted for deadenylases in the auxin degron strains might suggest an important role for deadenylation in cell growth. These seems to be a stronger effect than observed for *upf1*Δ mutants that block NMD.

In our hands and in the BY4741 genetic background, deletion of the genes for CCR4 and POP2 showed a strong growth defect, described in previous publications (for example . Traven et al., *Genetics* 2005, PMID 15466434, Traven et al., *Genetics* 2009, PMID 19487562 - a region of Fig. 5 of that paper is shown in **Fig. Reviewer 2/Answer6**). This effect is probably not specific to the S288C derived BY4741, as it was reported for the Sigma1278b strain as well (Lo et al., *Genetics* 2012 PMID 22595243).

Figure Reviewer 2/Answer 6

Deletion of CCR4 or POP2 affects yeast growth on YPD plates. Region of Fig. 5B from the Traven et al., *Genetics* 2009, article.

Thus, the slow growth phenotype of *S. cerevisiae* strains lacking CCR4 or POP2 seems to be robust. It was one of the major reasons why we developed the degron strains. When either Ccr4 or Pop2 were depleted, the degron strains grew poorly, similar to the deletion strains. This growth defect could be complemented by the expression of CCR4 or POP2 from a plasmid (new data presented in **Fig. EV3C**).

Deletion of UPF1, as well as that of other NMD factors, has little to no impact on the growth rate in *S. cerevisiae*. It is possible that other RNA degradation pathways compensate the lack of NMD.

Question 7: An increase in poly(A) tail length upon depletion of the Ccr4 and Caf1 deadenylases (3B, EV5) suggests that the mRNAs are subjected, at least to a degree, to initial deadenylation. If a deadenylation pathway is in a simple competition with a decapping-mediated pathway, then depletion of the deadenylases would stimulate decapping and the poly(A) tail lengthening effect would be dampened, particularly for unstable mRNAs with a high decapping rate. This data may therefore support a deadenylation-dependent initial phase within the pathway.

Answer 7: We agree with the reviewer that our Nanopore results showing a global increase in the poly(A) tail sizes for all RNAs when deadenylases were depleted, indicate that mRNAs are subjected to initial deadenylation. We do not fully agree, however, with the idea that deadenylation and decapping

are in competition. If the reviewer refers to the proposed uncoupling between deadenylation and initiation of decapping, the presence of mRNAs with longer poly(A) tails should not affect decapping. A different set of published results goes in the same direction, based on observations done in cells in which the Pan2/Pan3 deadenylase complex is inactive. In such cells, RNAs show a global increase in the poly(A) tail length (Brown et al., *Mol Cell Biol* 1996, PMID 8816488, Tudek et al., *Nat Commun* 2021, PMID 34400637) but RNA degradation rates are minimally affected (Sun et al., *Mol Cell* 2013, PMID 24119399).

Question 8: The data in 3C and 3D suggests that depletion of Dcp2, but not depletion of the deadenylases, causes a block in turnover (supported by data in Fig 5). If decapping rather than deadenylation were rate-limiting for an average mRNA, then I think this is what you would expect even if deadenylation preceded decapping.

Answer 8: We agree with the reviewer that it can be difficult to distinguish a situation in which deadenylation is required for decapping from the one in which the two are independent, when decapping is required in both models. However, the obtained results suggest that decapping activation and not deadenylation speed is the limiting step for RNA degradation. This is an important point that we tried to express in the manuscript because, in general, the current consensus is that deadenylation speed is the one that limits RNA degradation rates. Please note that we do not exclude the presence of deadenylation as an important part of RNA metabolism:

"We found that an increase in mRNA levels was accompanied by a decrease in the size of the poly(A) tails when Dcp2 was depleted (Fig. 3C). Such correlated changes were not observed in the strains depleted for the Ccr4 and Pop2 deadenylases (Fig. 3D), even though the poly(A) tail length was clearly and globally increased. Thus, perturbation of deadenylation or decapping led to results that are compatible with an uncoupling between these processes during the degradation of unstable RNAs in yeast." - page 6

Question 9: It would perhaps be insightful to deplete Dcp2 and the decapping enzymes together. This would address whether other pathways are involved to a significant degree.

Answer 9: We agree with the reviewer that depleting both deadenylases and decapping enzymes from yeast would potentially allow the identification of other RNA degradation pathways. However, we believe that this would represent an entirely new project that would not bring additional evidence to the main message of the current manuscript, that deadenylation rate does not seem to play the role that we thought it has as a limiting step in RNA degradation.

Question 10: It's not obvious why data is shown for the OPT-HIS3 and non-OPT-HIS3 reporters when the analysis of an unstable mRNA with a long poly(A) tail and a stable, house keeping transcript would have been more in keeping with other experiments shown.

Answer 10: A major advantage of the non-OPT-HIS3 and opt-HIS3 reporters is that, by design, we can block their transcription by adding doxycycline to the medium. We could thus follow the degradation kinetics for the reporters under conditions in which the cellular metabolism was minimally perturbed. Moreover, the relatively high levels of expression of these reporters allow the evaluation of the poly(A) tail dynamics by RNase H and Northern blot, which would be technically very challenging for an unstable endogenous transcript. Finally, similar reporters have been used to investigate the importance of the coding sequence in RNA stability, leading to several important new results about RNA degradation and the interplay between translation, the Ccr4-Not complex and decapping (eg Webster et al., *Mol Cell* 2018, PMID 29932902). In this publication, the OPT-HIS3 used was stabilized in the absence of Ccr4, shifting its half life from 16 to 43 minutes, but only marginally affected by Pop2 (Caf1) absence (shift from 16 to 19 min.). It is striking that the poly(A) profiles (**Fig. S7** of the Webster

et al. paper) for the degradation kinetics of OPT-HIS3 in an *rbp1-1* strain (block of transcription) clearly show the persistence of elongated poly(A) tails over time despite the minimal, or absent, effect on RNA half-life (**Fig. Reviewer 2, Answer 10**).

Figure Reviewer 2, Answer 10

Part of Fig. S7 of the Webster et al., paper, in which the poly(A) status of an OPT HIS3 reporter seems to be uncoupled from the deadenylation kinetics.

Question 11: It would be useful to show the wild-type poly(A) profiles for comparison in Fig EVC,D and stack the panels, as in Fig. 5G. I find the data in Fig EV5 quite informative as this data allows a direct comparison of poly(A) tail lengths during mRNA turnover. I would propose to include this in Fig 5 (Fig 5 panels A, B and F could be put in the EV figure).

Answer 11: We agree with the reviewer that visual results have to be presented in the main figures of the manuscript. We fully reorganized **Fig. 5** to show the results from previous Fig. EV5. The poly(A) profiles are now also shown stacked, as suggested. For convenience, an image of the new Fig. 5 is presented below (**Fig. Reviewer 2, Answer 11**):

We thank the reviewer for the comments, detailed questions and the critical analysis of the results and their implications.

Answers to questions from reviewer 3

Context for the answers and general considerations: The reviewer's comment begins by stating that our work "*explores the role of deadenylation in stimulating decapping*". This might be a slight misunderstanding, as we rather tested whether the rate of deadenylation could be considered as the main determinant of RNA degradation speed in yeast. We used mathematical modelling to understand why the published steady-state distribution of poly(A) tail lengths was so different from that "naively" expected from the original model of a mandatory deadenylation followed by degradation. We confirmed previous findings that used modeling for results from yeast (Cao et al., *RNA* 2001, PMID 11565744) or human cells (Eisen et al., *Mol Cell* 2020, PMID 31902669) showing that decapping rate must be highly variable among various transcripts with the same, short poly(A) tail, to explain experimental results. What we propose is a generalization of this model that includes the possibility that decapping can occur at any length of the poly(A) tail. Most importantly, we went further than the previously published work mentioned above, by experimentally testing the main prediction of our model, which is that mRNA degradation can occur at the same speed, even when the poly(A) tails of mRNAs are longer (when deadenylation was inactivated). We benefited from using yeast strains in which we could rapidly induce a deadenylation defect, in contrast to previous studies using slow growing strains. Thus, our main results do not come from modelling but from direct observations of the behaviour of poly(A) tails, mRNA levels and mRNA degradation kinetics.

In response to specific concerns raised by reviewer 3, please find below detailed explanations, additional experimental and data analysis results.

Question 1: *To conclude that "Pab1 preferentially binds long poly(A)-tail RNAs" needs more evidence. For instance, the authors need to show that RNAs bound by Pab1 have longer poly(A) tails than those originating from the same gene and not bound by Pab1. Given that this analysis and other hand-picked examples of gene classes are the bulwark of their conclusion that "the association of an mRNA with Pab1 can be considered a marker of their instability," substantially more experiments need to be performed to exclude other alternative explanations.*

Answer 1: We agree with the reviewer that the original version of the manuscript lacked this control. The experimental evidence that the RNAs co-purified with Pab1-TAP have long poly(A) tails is shown in **Fig. EV2D** of the revised manuscript (please see the explanations and **figure** in response to **Question 2 of reviewer 2**). We used ePAT, a method that allows the estimation of poly(A) tail distribution with high sensitivity. Directly answering the reviewer question, the fraction of an endogenous transcript that was bound by Pab1 showed longer poly(A) tails as compared to the input RNA.

In addition to the experimental evidence that answers the reviewer's question, we would like to point out that the "*hand-picked examples of gene classes*" mentioned by the reviewer represent two hundred different XUT/SUT transcripts that are unstable, have longer than average poly(A) tails, and for which RNA sequencing provided a sufficient number of reads for quantitation in the input and purified fractions. These transcripts are degraded by nonsense-mediated mRNA decay, without prior deadenylation. We are in the privileged situation in *S. cerevisiae* of being able to use an entire class of diverse unstable transcripts to analyze experimental results. We believe that, together with the data presented in **Fig. EV2D**, this answers the reviewer's first question and clarifies our choice of data analysis presented in the manuscript.

Question 2: *The authors mention Eisen, et al 2020, but do not discuss the fundamental differences in their models from the published one: here, the authors assume that decapping rate is constant with*

different poly(A)-tail lengths (for transcripts from the same gene), which is opposite to the conclusions of Eisen et al. (which found that the two rates, in mammalian cells, are often strongly correlated and decapping rates are not constant). This assumption appears to be the basis of the difference in conclusions between the two papers. Indeed, the previously published paper explained the lack of correlation between tail lengths and stability to be partly due to this fundamental link between deadenylation and decapping rates. How do the authors' models and conclusions change if they do not assume the same decapping rates for different tail lengths? How can the authors exclude this alternative model?

Answer 2: We agree that the original version of the manuscript did not include a detailed comparison with previous models, such as the one used by Eisen et al. The short answer to the question of whether we have considered the "classical" model of RNA degradation, including a highly variable decapping step occurring mainly at short poly(A) tails, similar to the one used by Eisen et al., to explain our results is "yes".

The reviewer asked whether we could exclude an alternative model in which decapping rates would be different at different poly(A) tail lengths for a given transcript (increasing with decreased length). The model used in Eisen et al. assumed that the probability of an mRNA being degraded increases as the size of its poly(A) tail decreases, and that this probability follows a logistic function. For 97% of the transcripts analyzed, they found that decapping (or degradation) occurred mostly at a mean tail length of 50 nucleotides, or 25 nucleotides (for 69% of transcripts). Thus, in reality, the model used is similar, albeit more complex, to the one we presented in **Fig. 2C** of our manuscript, under the label "deadenylation-dependent decapping" (copied here in **Fig. Reviewer 3 , Answer 2, A** for convenience). We used this model to show that it can only explain the steady-state distribution of yeast poly(A) tails as a function of mRNA stability if the decapping rates are highly variable, high for unstable transcripts and low for stable ones. We have therefore carried out an analysis that is similar to the one requested. What is really important, and we believe this concept is not yet clear in the RNA community, is that for a similar deadenylation rate, it is the decapping speed that would dictate the half-life of an mRNA.

Figure Reviewer 3, Answer 2. Deadenylation-dependent model for RNA degradation and parameter correlations in previous work.

(A) The model we used in the manuscript to estimate the expected steady-state relationship between mRNA half-life and average poly(A) tail length.

(B) Correlations between RNA production and degradation parameters estimated in the Eisen et al., *Mol Cell* 2020 paper. Data were extracted from Table S1 (cell line 2). The numbers represent Pearson correlation coefficients for log₂ transformed reported values (3140 transcripts).

The reviewer mentioned the strong correlation between deadenylation and decapping rates estimated by Eisen et al., for human transcripts. We examined the transcript-specific parameters that were estimated from the pulse-labelling experiments with 5-ethynyl-uridine in that paper. A strong correlation was found between deadenylation and decapping rates (Pearson coefficient of 0.6), as mentioned by the reviewer (**Fig. Reviewer 3, Answer 2, B**). However, strong correlations were also present between the estimated production rate of the transcript and the deadenylation and decapping rates (0.49 and 0.51). Furthermore, a strong **negative** correlation was found between the production rate (including synthesis, splicing and nuclear export of mRNA) and the half-life of the analysed transcripts. This finding is curious, as there is no known reason why the most stable transcripts (estimated to have long half-lives) would have the lowest production rates. Previous estimates of production rates and mRNA stability, based on metabolic labelling did not show such a trend (Schwanhäusser et al., *Nature* 2011, PMID 21593866, correlation coefficient of 0.04 between transcription rate and half-life, 4336 transcripts). We believe that, as a community, we should be very cautious about attributing biological significance to results based on correlations between parameters of a model. Either the situation is different among different organisms and experimental systems, or the Eisen et al., model is not entirely appropriate for analysing the results, even if it is perfectly capable of modeling them. Thus, the finding that deadenylation and decapping rates are correlated may need to be independently validated.

The role of modelling is both descriptive, to find parameters that fit the data, and predictive. We may not have emphasised enough in our manuscript the main reason why we favour a model in which decapping can occur at any poly(A) tail length, as opposed to the classical "deadenylate and then decap" model. Please note that the conclusions from Eisen et al., model about the correlation between deadenylation and decapping speeds were not experimentally validated by perturbing deadenylation or decapping rates. In this respect, we performed experiments, in which we rapidly inactivated decapping or deadenylation and examined mRNA decay both on a large scale using SLAM-Seq, and on a specific reporter. Our results provide evidence, independent of any model or particular assumptions, suggesting that deadenylation and decapping are uncoupled processes that can occur independently and at various speeds.

We agree that an extended discussion of similarities and differences with the Eisen et al., model is useful for our manuscript and we increased the size of the corresponding paragraph in the Discussion, copied here for convenience:

*"Our results might seem at odds with the widely accepted idea that deadenylation precedes decapping in the degradation of RNA in eukaryotes. Still, recent results have already suggested that deadenylation is not linked with RNA degradation, at least during yeast meiosis (Wiener et al, 2021). Even the results of large-scale metabolic labeling and poly(A) tail length measurements in 3T3 mouse cells (Eisen et al, 2020) are compatible with a deadenylation-independent mechanism for RNA degradation. Individual highly unstable transcripts, such as *Metrn1*, showed a broad distribution of poly(A) tails at steady-state, while a very stable one, such as *Eef2*, displayed a more compact distribution, with shorter poly(A) tails on average. Individual cases were not reflected in a global correlation between poly(A) tail length and RNA stability, potentially because deadenylation and decapping have different ranges and dynamics in mammalian cells compared with yeast. Importantly, fitting a deadenylation-dependent model for RNA degradation involved a highly variable parameter, the speed of decapping for oligoadenylated RNA species. This speed varied by a factor of 1000 (Eisen et al, 2020), which is similar to the range of variation in the reported model for the deadenylation speed. Thus, the published model only fits experimental data if it includes a highly variable and limiting decapping speed. A strong correlation between measured deadenylation and decapping rates (Pearson correlation coefficient of 0.6, based on data presented in*

Table S1, 3140 transcripts) might indicate that unstable mRNA are particularly prone to deadenylation and that, once they reach a poly(A) threshold get decapped also rapidly. This result, based on the correlation of the parameters obtained by data fitting to a specific model, need to be critically assessed. For example, further examination of the reported parameters of the degradation kinetics of thousands of transcripts showed a strong positive correlation between deadenylation rate and RNA production rate (Pearson correlation coefficient 0.49) and also between decapping and RNA production (Pearson correlation coefficient of 0.51). These results would imply that the most highly produced RNAs have also the fastest deadenylation and decapping rates, a phenomenon that was not observed previously in mammalian cells (Schwanhäusser et al, 2011) and has no obvious biological meaning. This example illustrates why observed correlations between parameters of a model could be misleading, unless they can be independently validated."

Question 3: The authors deplete various deadenylase components that could also impact the deadenylation-decapping coupling implied by Eisen et al, papers from the Izaurralde group, etc. What happens when catalytic mutants are added back? Can the authors distinguish between the catalytic aspect of deadenylation vs a structural requirement for the deadenylase complexes?

Answer 3: The short answer to this question is that we cannot complement a phenotype (stabilization of mRNA) that is not present. For example, our reporter mRNA showed no change in its half-life when the two major yeast deadenylases were depleted. However, it clearly showed longer poly(A) tails (original Fig. EV5, panel A, now Fig. 5E copied here for convenience).

Figure 5E. Slowing down deadenylation does not change the half-life of a reporter mRNA. RNase H experiments showing the size of the poly(A) tails at steady-state (time 0) or at different time points after inhibition of reporter transcription by doxycyclin addition.

We agree with the reviewer that the presence of proteins alone, independent of their deadenylase activity, may be required for specific phenotypes. Our manuscript cited a recently published example of the effect of the Ccr4-Not complex on mRNA through ribosome ubiquitination rather than via mRNA deadenylation (Absmeier et al., *Nat Struct Mol Biol* 2023 PMID 37653243). Since we observed no change in mRNA decay in the deadenylation enzyme depleted strains, we do not dispose of an mRNA degradation phenotype that can be tested with either catalytically active or inactive enzymes. Please also note that, as explained in the answer to the previous question, the correlation between deadenylation and decapping rates suggested in the Eisen et al., paper, is not well established and has not been fully and experimentally validated.

The reviewer mentioned papers from the Izaurralde group, which has made many very important contributions to the understanding of RNA metabolism. We looked for specific papers that showed how deadenylation and decapping might be connected. It is possible that the reviewer is referring to a

paper published in 2010 (Haas et al, *J Cell Biol* 2010, PMID 20404111), which found a link between deadenylation and decapping in *Drosophila melanogaster* and proposed that this link was mediated by the Pat1(HPat)/Lsm1-7 complex. Our extensive quantitative mass spectrometry results performed with yeast Pat1, Lsm1, Lsm7 and Dhh1 (**Fig EV1**) identified a strong enrichment of decapping factors together with yeast Pat1/Lsm1-7, but we did not find any interaction with members of the Ccr4-Not complex. In previous experiments we also identified the proteins associated with the decapping complex, in particular Dcp1, and also found no evidence of interaction with the Ccr4-Not complex (Dehecq et al., *EMBO J*, 2018 PMID 30275269). We also looked at the available results from high-throughput mass-spectrometry analyses of purified complexes in mammalian cells. While components of the Ccr4-Not complex were identified in these experiments, no interaction with decapping or Pat1/Lsm proteins was detected in mouse cells (Hein et al., *Cell* 2015 PMID 26496610) or human cells in culture (Huttlin et al., *Cell* 2021, PMID 33961781, data can be explored at bioplex.hms.harvard.edu).

It is therefore unclear whether the *Drosophila* results apply to yeast or mammalian RNA metabolism and the potential link between deadenylation and decapping machineries remains to be investigated.

In conclusion, we believe that the current literature and our biochemical results do not strongly support previous claims of a biochemical link between decapping and deadenylation in *S. cerevisiae*. Moreover, such a link is not required to understand the distribution of poly(A) tails in relation with mRNA stability. Furthermore, since we do not observe an RNA degradation phenotype associated with deadenylases depletion, even though we have measured a clear and global change in poly(A) tail lengths under these conditions (e.g. **Fig. 3B**), there is no phenotype that we can challenge or test with catalytically inactive deadenylases.

As a further control, and following the reviewer's comments, we now provide additional experimental results obtained by using plasmids carrying catalytically active or mutated versions of CCR4 and POP2. These complementation experiments showed that the degron strains recovered their growth rates when the plasmids were present (new **Fig. EV3C**). Moreover, the extended poly(A) phenotype for an endogenous transcript, measured by ePAT when either Ccr4 or Pop2 were depleted, was complemented by the wild type version of Ccr4, but not by a Ccr4(E556A) variant. For Pop2 the situation was different: both the wild type and the D310A mutant versions of the protein were able to complement the growth defect and the extended poly(A) phenotype. These results are described in a new paragraph included in the revised manuscript:

"The growth defect of degron strains could be reverted in the presence of a plasmid carrying the corresponding deadenylase (Fig. EV3C). Expression of either CCR4 or POP2 could not complement the growth defect of the double degron strain, as expected (Fig. EV3D). As a further validation of these complementation results, we tested the poly(A) status of an endogenous transcript, HHT2, by ePAT. We observed that the increase in poly(A) length observed when Ccr4 was depleted could be reverted when CCR4 was expressed, but not when a catalytically inactive version of the enzyme, E556A (Chen et al, 2002), was used (Fig. EV3E, F). The shift in poly(A) tail length for HHT2 was more modest when POP2 was depleted, and the Pop2(D310A) mutant, supposed to affect the enzymatic activity of Pop2, was as effective as the wild type in reverting the poly(A) and growth phenotype (Fig. EV3C, E, G). It is possible that the D310A change is not sufficient to completely inactivate the deadenylation activity of Pop2, as demonstrated *in vitro* (Ye et al, 2021). Alternatively, the presence of the protein is sufficient to activate the deadenylase activity of Ccr4, as previously suggested (Tucker et al, 2002). In conclusion, the degron system allows rapid depletion of the major deadenylases, with an impact on the poly(A) length of an endogenous transcript."

The corresponding panels are copied here for convenience:

Portion of Fig. EV3 of the revised manuscript. (C) Serial dilution plate growth assay for deletion and inducible degron (ideg) strains affecting deadenylases of the CCR4/NOT complex. Growth on rich medium (YPD, left) was compared with growth on rich medium containing β -estradiol and IAA (right). The strains depleted for Ccr4 or Pop2 were transformed with a control plasmid (pRS315), pRS315-CCR4 or pRS315-POP2. The asterisks indicates mutated versions of Ccr4, E556A, and Pop2, D310A. **(D)** Similar with (C), but for the concomitant depletion of Ccr4 and Pop2. **(E)** Changes in the poly(A) tail size of HHT2 measured using ePAT in the strains presented in panel (C). The TVN-PAT lane indicates the relative position of the A12 RNA. **(F)** Poly(A) size profile for HHT2 under Ccr4 depletion conditions was extracted from panel (E) and normalized to the signal in each lane. **(G)** Similar to (F), but for the depletion of Pop2.

We thank the reviewer for the critical analysis of our manuscript and we would be happy to engage in constructive conversations over any specific points the editor or the reviewers might want further clarifications.

Dear Dr. Saveanu,

Thank you for submitting a revised version of your manuscript. Your study has now been seen by all original referees, who find that their previous concerns have been addressed and now recommend publication of the manuscript. As you can see from the comments referee #3 thinks that the manuscript might benefit from additional discussion of drawbacks and caveats of their model. Otherwise, there remain only a few mainly editorial points that have to be addressed before I can extend formal acceptance of the manuscript:

- As we are switching from a free-text author contribution statement towards a more formal statement based on Contributor Role Taxonomy (CRediT) terms, please remove the present Author Contribution section and instead specify each author's contribution(s) directly in the Author Information page of our submission system during upload of the final manuscript. See <https://casrai.org/credit/> for more information.
- Please adjust the in-text callouts for individual figures and figure panels: e.g. Fig. S2A, B and Fig. S2C, D should be corrected to Appendix Fig. S2xx; callouts for Table S1 should be corrected to Appendix Table S1, and also include the missing callouts for Appendix Table S2-S3
- Please complete the Figure Source checklist (general info table not completed)
- Please place the Appendix table legends above the corresponding tables and include page numbers of the listed items in the table of contents (ToC);
- EXPANDED VIEW DATASETS (call-out: "Dataset EV1/2/3/4"), and the spreadsheets each need a separate "Legend" tab containing dataset title and legend information, that should be removed from the Table of Contents in Appendix PDF
- Please provide the Reagent and Tools Table. For more information, please check <https://www.embopress.org/page/journal/14602075/authorguide#structuredmethods> and download the template for Reagent Table (attached for your convenience)
- Please provide suggestions for a short 'blurb' text prefacing and summing up the conceptual aspect of the study in two sentences (max. 250 characters), followed by 3-5 one-sentence 'bullet points' with brief factual statements of key results of the paper; they will form the basis of an editor-written 'Synopsis' accompanying the online version of the article. Please also provide an altered synopsis image, making sure that the aspect ratio conforms to our website's format - it should be exactly 550 pixels wide and between 300-600 pixels high.
- please note that the valid and accessible URLs for GSE160642, GSE247954, and GSE211782 datasets should be provided in the data availability statement.
- Please remove the "Appendix Material" section from ms file (including the legends for Fig. S1-S3 and Datasets EV1-EV4)
- Finally, there are a couple of points in the Figure Legends (main + EV) raised by our data editors:
 1. Please note that the exact p values are not provided in the legends of figures 1c; 3b; EV 2a.
 2. Please note that information related to n is missing in the legend of figure EV 1e.
 3. Please note that the error bars are not defined in the legend of figure EV 1e.

With best regards,

Cornelius Schneider

Referee #1:

I am satisfied with the author's responses to the other reviewers. I believe this process has significantly improved the clarity of the manuscript.

Referee #2:

This manuscript is a very solid study that provides significant novel insights into the processes of mRNA turnover in yeast. Through modelling of mRNA turnover kinetics, rapid genetic depletion of deadenylation and decapping enzymes and the analysis of mRNAs on a transcriptome-wide basis and through the analysis of specific reporter transcripts, the data strongly support the central conclusion that decapping and degradation of mRNA before extensive deadenylation is a widespread phenomenon. These findings are contrary to the general consensus model of euakryotic mRNA turnover and will provide a basis to reevaluate existing data and address in more detail the mechanistic basis of this process in future.

The authors have appropriately addressed the comments raised in my initial review and I support publication of the revised manuscript.

Referee #3:

I commend the authors on their thoughtful revision and response to reviews. Although I still am not entirely convinced by some interpretations, I think these differences reflect the current open questions in the field and a difference in relative weighting of published results; in fact, I would suggest that the ideas and questions sparked by the authors' results highlight the importance of their manuscript, rather than diminishing it. My only minor comment is that further potential limitations or caveats of their models, as well as alternative explanations, in the Discussion would be a welcome addition for the reader. I support publication of this manuscript.

All editorial and formatting issues were resolved by the authors.

Dear Dr. Saveanu,

I am pleased to inform you that your manuscript has been accepted for publication in the EMBO Journal.

Yours sincerely,

Cornelius Schneider, PhD
Editor
The EMBO Journal
c.schneider@embojournal.org
